# Leveraging Sparsity for Sample-Efficient Preference Learning: A Theoretical Perspective

Yunzhen Yao [1]   Lie He [2 3]   Michael Gastpar [1]

## Abstract

This paper considers the sample-efficiency of preference learning, which models and predicts human choices based on comparative judgments. The minimax optimal estimation error rate $\Theta(d/n)$ in classical estimation theory requires that the number of samples $n$ scales linearly with the dimensionality of the feature space $d$. However, the high dimensionality of the feature space and the high cost of collecting human-annotated data challenge the efficiency of traditional estimation methods. To remedy this, we leverage sparsity in the preference model and establish sharp error rates. We show that under the sparse random utility model, where the parameter of the reward function is $k$-sparse, the minimax optimal rate can be reduced to $\Theta(k/n \log(d/k))$. Furthermore, we analyze the $\ell_1$-regularized estimator and show that it achieves near-optimal rate under mild assumptions on the Gram matrix. Experiments on synthetic data and LLM alignment data validate our theoretical findings, showing that sparsity-aware methods significantly reduce sample complexity and improve prediction accuracy.

## 1. Introduction

### 1.1. Motivation

Preference learning focuses on modeling and predicting subjective choices or priorities from empirical comparative data to support tasks such as decision-making, ranking, and recommendation. For example, commercial recommender systems select items from a set of candidates based on user preferences (Resnick & Varian, 1997; Rendle et al., 2009; He et al., 2017). Similarly, information retrieval systems can leverage user clickthrough data from search-engine query logs to improve the relevance of retrieved results (Joachims, 2002; Burges et al., 2005; Liu et al., 2009). More recently, large language models (LLMs) are often pretrained on large-scale internet data, which may contain harmful or biased content, making direct deployment risky. Learning from human preferences is thus adopted to align pretrained models with human values and objectives (Christiano et al., 2017; Stiennon et al., 2020; Ouyang et al., 2022; Bai et al., 2022).

Preferences among alternatives can be represented by a real-valued *utility function*, where a higher function value corresponds to a more preferred option, provided that the preference relation is complete, transitive, and continuous, according to Debreu's representation theorem (Debreu et al., 1954). Moreover, to account for inconsistencies and randomness in human decision-making—arising from subjective interpretations, ambiguous guidelines, and fluctuating focus—a deterministic utility function can be extended to a *stochastic* utility model, in which the probability of choosing one alternative is higher when its utility is greater. This paper focuses on learning preferences by training a parameterized random utility model.

A major challenge in preference learning is the high cost of collecting human preference data (Gao et al., 2023; Wang et al., 2023; Mahan et al., 2024). For example, aligning LLMs with human values requires a significant amount of samples labeled by experienced human annotators who select the most "helpful" and "harmless" response among all candidates. This difficulty is compounded by the fact that alternatives often lie in feature spaces whose dimensionality $d$ far exceeds the number of available samples $n$, leading to high estimation error $\Theta(d/n)$ for prevalent maximum likelihood methods (Shah et al., 2016; Faury et al., 2020; Saha et al., 2023; Zhu et al., 2023).

While alternatives may have thousands of attributes (dimensions), human preferences are usually driven by only a small set of critical factors in a given context. For instance, when a user selects among smartphones, the decision might hinge primarily on price, camera quality, and UI design, whereas many other attributes (e.g., place of manufacture)

[1] LINX, EPFL, Lausanne, Switzerland [2] Key Laboratory of Interdisciplinary Research of Computation and Economics (Shanghai University of Finance and Economics), Ministry of Education, China [3] School of Computing and Artificial Intelligence, Shanghai University of Finance and Economics, Shanghai, China. Correspondence to: Yunzhen Yao <yunzhen.yao@epfl.ch>.

*Proceedings of the 42nd International Conference on Machine Learning*, Vancouver, Canada. PMLR 267, 2025. Copyright 2025 by the author(s).

*Table 1.* Estimation error rates for preference learning in non-sparse and sparse settings with error metric $\| \cdot \|_\Sigma^2$. Notation: $d$ is the ambient dimension, $k$ is the sparsity level, and $n$ is the sample size.

| | | | |
|---|---|---|---|
| Non-Sparse Settings | Minimax Optimal | $\Theta\left(\frac{d}{n}\right)$ [†] | |
| Sparse Settings | Minimax Optimal | $\Theta\left(\frac{k\log(d/k)}{n}\right)$ | Theorem 3.1 and 3.2 |
| | $\ell_1$-Regularized (Slow) | $\mathcal{O}\left(\sqrt{\frac{k\log d}{n}}\right)$ | Theorem 3.3 |
| | $\ell_1$-Regularized (Fast) | $\mathcal{O}\left(\frac{k\log d}{n}\right)$ | Theorem 3.4 |

[†] (Shah et al., 2016; Faury et al., 2020; Saha et al., 2023; Zhu et al., 2023).

may have little influence for that user. Similarly, a reader's preference over articles may depend solely on the presence of a few key words. When the feature vector is a binary indicator over a large vocabulary, the reward parameter is naturally sparse. Moreover, in many modern applications, such as LLM alignment or recommendation systems, feature spaces often contain thousands to millions of dimensions, rendering identifying relevant features beforehand impractical. The concept of *sparsity* offers a promising way to address the above challenges. Building on this idea, the well-established field of *compressed sensing* demonstrates how leveraging sparsity can significantly reduce sample complexity (Donoho, 2006; Candes & Tao, 2006; Tropp & Gilbert, 2007; Ye & Zhang, 2010; Rigollet & Tsybakov, 2011; Raskutti et al., 2011; Verzelen, 2012; Candes & Davenport, 2013), making sparsity-aware approaches particularly promising for preference learning. Despite successes in other domains, the theoretical and empirical foundations of sparsity in preference learning remain underdeveloped, pointing to a rich area for further study.

## 1.2. Contribution

In this paper, we focus on the problem of sample-efficient estimation for preference learning models. Since the sample complexity scales linearly with the ambient dimension (Shah et al., 2016; Faury et al., 2020; Saha et al., 2023; Zhu et al., 2023), the high dimensionality of the ambient space poses a bottleneck for accurate estimation. To address this challenge, we consider the *sparse RUM* setting (see Equation (5) below), where the model parameter is $k$-sparse. Under this assumption, we show that the upper and lower bounds on estimation error rates can be improved with respect to $d$. To the best of our knowledge, this work is the first to theoretically investigate sparsity in preference learning and analyze its impact on estimation rates. Specifically, our contributions are as follows.

- **Minimax lower bound.** We establish an information-

theoretical lower bound of $\Omega\left((k/n)\log(d/k)\right)$ for the empirical estimation error in the sparse RUM setting, contrasting it with $\Omega(d/n)$ in the non-sparse setting.

- **Minimax optimal rate.** We show that an $\ell_0$-constrained estimator achieves the minimax-optimal rate under the common strong log-concavity assumption that covers a class of popular models like Bradley-Terry-Luce (BTL) model (Bradley & Terry, 1952; Luce, 1959) and Thurstone-Mosteller (TM) model (Thurstone, 1994; Mosteller, 2006).

- **Upper bounds for the $\ell_1$-regularized estimator.** We show that, with a penalty of $\Theta(1/\sqrt{n})$, the $\ell_1$-regularized estimator achieves the rate $\mathcal{O}\left(\sqrt{(k/n)\log d}\right)$. Furthermore, under certain assumption on the spectrum of the Gram matrix, it attains a sharper rate $\mathcal{O}\left((k/n)\log d\right)$, which is nearly minimax optimal.

- **Experimental evaluation.** Our experimental evaluations demonstrate that sparsity-aware estimators outperform widely used baselines in reward modeling, evaluated on both synthetic datasets and LLM alignment datasets using popular language models.[1] These findings underscore the potential of sparsity-aware approaches in preference-based tasks, including reinforcement learning from human feedback (RLHF).

We summarize the estimation error rates across different settings in Table 1. In contrast to classical regression problems, which rely on cardinal labels of measurable quantities, preference learning only has access to pairwise comparison data, each providing at most one bit of information. Despite these challenges, our upper and lower bounds on estimation errors remain in the same order as those in classical

---

[1]Code can be found at this link: `https://github.com/yaoyzh/SparsePreferenceLearning`

compressed sensing (Donoho, 2006; Candes & Tao, 2006; Tropp & Gilbert, 2007).

# 2. Preliminaries

## 2.1. Problem Formulation of Preference Learning

Let $\mathcal{A}$ be a set of alternatives, and let $\phi : \mathcal{A} \to \mathbb{R}^d$ be a fixed and known feature map, where $\phi(a)$ represents a $d$-dimensional feature vector corresponding to $a \in \mathcal{A}$. The feature space of $\mathcal{A}$ induced by $\phi$ is defined as the image of $\phi$, denoted as $\mathcal{D} := \phi(\mathcal{A}) \subset \mathbb{R}^d$. A preference relation defined on $\mathcal{D}$ satisfies Debreu's representation theorem can be characterized by a reward or utility function $r^*$. Specifically, for two feature vectors $x_0, x_1 \in \mathcal{D}$ such that $r^*(x_0) < r^*(x_1)$, we state that $x_1$ is *preferred* to $x_0$, denoted as $x_0 \prec x_1$. The ground-truth reward function $r^*$ is fixed and unknown. We assume that the feature map $\phi$ on $\mathcal{A}$ accounts for the non-linearity, whereas $r^*$ is linear.

Consider a preference dataset comprising $n$ pairs of samples drawn from $\mathcal{D}$, denoted as $\{\xi_i\}_{i=1}^n$, where each sample $\xi_i$ is represented as

$$\xi_i = (x_{0,i}, x_{1,i}, y_i) \in \mathcal{D} \times \mathcal{D} \times \{0, 1\}.$$

Here, $x_{0,i} := \phi(a_{0,i})$ and $x_{1,i} := \phi(a_{1,i})$ are the feature vectors of the alternatives being compared. The binary variable $y_i$ is the preference signal, with $y_i = 0$ indicating $x_{0,i}$ is *observed* to be preferred over $x_{1,i}$, and $y_i = 1$ indicating the opposite. In this paper, we consider a fixed design setup, where $\{(x_{0,i}, x_{1,i})\}_{i=1}^n$ is *deterministic*, and $\{y_i\}_{i=1}^n$ is the realization of the set of random variables $\{Y_i\}_{i=1}^n$. Specifically, $Y_i$ conforms to the *random utility model*.

**Random utility model (RUM).** To account for potential inconsistencies or randomness in human decision-making, the random utility model assumes that the probability of choosing $x_0$ is higher than choosing $x_1$ if $r^*(x_0)$ is greater than $r^*(x_1)$. Specifically, the conditional distribution $P_{Y|(X_0, X_1)}$ is

$$P(Y = 0 \mid x_0, x_1) = F\left(\frac{r^*(x_0) - r^*(x_1)}{\sigma}\right) \quad (1)$$

where $F : \mathbb{R} \to [0, 1]$ satisfies $F(t) = 1 - F(-t)$, and $\sigma \in \mathbb{R}_+$ is the randomness level of $Y$. If $F(t)$ is the sigmoid function, i.e., $F(t) = \frac{1}{1+\exp(-t)}$, then (1) corresponds to the well-known Bradley-Terry-Luce (BTL) model (Bradley & Terry, 1952; Luce, 1959). If $F(t)$ is the cumulative distribution function of the standard Gaussian distribution, then (1) becomes the Thurstone-Mosteller (TM) model (Thurstone, 1994; Mosteller, 2006).

We assume $r^* : \mathcal{D} \to \mathbb{R}$ is parameterized by $\theta^* \in \mathbb{R}^d$, i.e.,

$$r^*(x) = \langle \theta^*, x \rangle. \quad (2)$$

The goal of *preference learning* is to estimate the parameter $\theta^*$ of the reward function $r^*$, based on preference samples $\{\xi_i\}_{i=1}^n$.

**Maximum likelihood (ML) estimator.** Given $n$ samples $\{\xi_i\}_{i=1}^n$, the *negative log-likelihood* for a parameter $\theta \in \mathbb{R}^d$ is defined as

$$\mathcal{L}(\theta; \{\xi_i\}_{i=1}^n) := -\frac{1}{n} \sum_{i=1}^n \log F\left((-1)^{y_i} \frac{\langle \theta, x_{0,i} \rangle - \langle \theta, x_{1,i} \rangle}{\sigma}\right)$$

We suppose that $\theta^*$ is bounded by a constant, i.e.,

$$\theta^* \in \Theta := \{\theta \in \mathbb{R}^d : \|\theta\|_2 \leq B\}.$$

The maximum likelihood (ML) estimator $\hat{\theta}_{\mathrm{ML}}$ is defined as

$$\hat{\theta}_{\mathrm{ML}} \in \arg\min_{\theta \in \Theta} \mathcal{L}(\theta, \{\xi_i\}_{i=1}^n). \quad (3)$$

**Performance measure.** We measure the performance of an estimate $\hat{\theta}$ using the *empirical error*, defined as

$$\frac{1}{n} \sum_{i=1}^n \left((\hat{r}(x_{0,i}) - \hat{r}(x_{1,i})) - (r^*(x_{0,i}) - r^*(x_{1,i}))\right)^2$$
$$= \frac{1}{n} \sum_{i=1}^n \left\langle \hat{\theta} - \theta^*, x_{0,i} - x_{1,i} \right\rangle^2$$

where $\hat{r}$ is the estimated reward function associated with $\hat{\theta}$. The *Gram matrix* $\Sigma \in \mathbb{R}^{d \times d}$, also called the *data covariance matrix*, associated with

$$X := \left[(x_{0,1} - x_{1,1}), \cdots, (x_{0,n} - x_{1,n})\right]^\top \in \mathbb{R}^{n \times d},$$

is defined by

$$\Sigma := \frac{1}{n} X^\top X = \frac{1}{n} \sum_{i=1}^n (x_{0,i} - x_{1,i})(x_{0,i} - x_{1,i})^\top$$

The Gram matrix induces a semi-norm

$$\|\theta\|_\Sigma := \sqrt{\theta^\top \Sigma \theta}, \quad \theta \in \mathbb{R}^d,$$

often called the *data-induced semi-norm*. The empirical error is the estimation error in the squared data-induced semi-norm, i.e.,

$$\left\|\hat{\theta} - \theta^*\right\|_\Sigma^2 = \frac{1}{n} \sum_{i=1}^n \left\langle \hat{\theta} - \theta^*, x_{0,i} - x_{1,i} \right\rangle^2. \quad (4)$$

Evaluating the estimation error using the squared data-induced semi-norm yields estimation error rates independent of data distribution.

In the rest of the paper, we use the term *reward*, while noting *utility* is often used interchangeably in related contexts. The complete list of notations can be found in Appendix B.

## 2.2. Preference Learning and RLHF

Preference learning serves as a foundational component of *Reinforcement Learning with Human Feedback* (RLHF). Here we focus on reward-based RLHF.

A preparatory step for RLHF is supervised fine-tuning (SFT), which fine-tunes the pretrained model on high-quality demonstration data, enabling the model to mimic the provided examples (e.g., summarizations or dialogues).

Next, RLHF aligns the model's behavior with human preferences by using human feedback. Let $\mathcal{S}$ denote the prompt (state) space and $\mathcal{A}$ represent the set of responses (actions or alternatives). For reward-based RLHF, the first step is to train a reward model to approximate the unknown reward function reflecting human preferences from the human preference data $\{s_i, a_{0,i}, a_{1,i}, y_i\}_i$, where $y_i \in \{0, 1\}$ indicates whether $a_{0,i}$ or $a_{1,i}$ is chosen as the preferred response given prompt $s_i$. This step is called *reward modeling*, and is exactly the problem of preference learning formulated in Section 2.1. To be specific, let $\phi$ be a known and fixed [2] feature mapping $\phi(s, a) : \mathcal{S} \times \mathcal{A} \to \mathbb{R}^d$, typically a language model with the last layer removed. For a given prompt $s_i \in \mathcal{S}$, the image $\phi(s_i, \mathcal{A})$ is the feature space $\mathcal{D}$ in preference learning. With the feature map $\phi$, the preference data can be represented as $\{\phi(s_i, a_{0,i}), \phi(s_i, a_{1,i}), y_i\}_i$.

Once a reward model is trained, the remaining step of RLHF is to further fine-tune the supervised fine-tuned (SFT) model using reinforcement learning (RL) algorithms, leveraging the reward model to optimize the policy for better alignment with human preferences and objectives.

In this work, we focus exclusively on preference learning (reward modeling), leaving other components of the RLHF process, including SFT and RL, unmodified.

# 3. Theoretical Foundations of Sparse Preference Learning

We propose *sparse preference learning*, wherein the ground-truth parameter $\theta^*$ in Equation (2), and accordingly the RUM framework in (1), is $k$-sparse, with $k$ potentially unknown. Formally, we consider the *sparse RUM* as follows.

---

**Sparse RUM**

$$P(Y = 0 \mid x_0, x_1) = F\left(\frac{\langle \theta^*, x_0 \rangle - \langle \theta^*, x_1 \rangle}{\sigma}\right)$$

$$\theta^* \in \Theta_{B,k} := \{\theta \in \mathbb{R}^d : \|\theta\|_2 \le B, \ \|\theta\|_0 \le k\}. \quad (5)$$

---

Furthermore, throughout the paper, we assume the feature

---

[2]We make this assumption for simplicity, while in Ouyang et al. (2022), $\phi$ is also trainable.

space $\mathcal{D}$ is bounded, i.e., there is a constant $L > 0$ such that

$$\|x_1 - x_2\|_2 \le L, \quad \forall x_1, x_2 \in \mathcal{D}. \quad (6)$$

The parameters $B$ and $L$, along with the function $F$ and the randomness level $\sigma$, determines a parameter $\zeta$ defined as

$$\zeta := \frac{\max_{t \in [0, BL/\sigma]} \ (F'(t))^2}{F(BL/\sigma)\,(1 - F(BL/\sigma))} \quad (7)$$

We observe that when $B = L = \sigma = 1$, the parameter $\zeta = 1.99$ in the BTL model and $1.19$ in the TM model, respectively. In practical scenarios, since the problem-dependent parameters $\sigma$ and $\zeta$ are generally independent of $d$ and $n$, the parameters $B, L, \sigma$, and $\zeta$ have $\mathcal{O}(1)$ values (Negahban et al., 2017; Shah et al., 2016). We thus consider these parameters to remain fixed.

In Section 3.1, we establishes an information-theoretical lower bound for sparse preference learning. In Section 3.2.1, we demonstrates that the $\ell_0$-constrained estimator achieves the minimax optimal rate. In Section 3.2.2, we provides two estimation error rates for the $\ell_1$-regularized estimator under difference assumptions. All the proofs are presented in Appendix E.

## 3.1. Minimax Lower Bound

To characterize the fundamental limits of sparse preference learning, Theorem 3.1 establishes a minimax lower bound for the empirical error (4).

**Theorem 3.1** (Minimax lower bounds). *Consider the sparse RUM (5) with $k \le rank(\Sigma)/8$. For a sample size*

$$n \ge \frac{\sigma^2}{64B^2\zeta\lambda_{rank(\Sigma)}} k \log\left(1 + \frac{rank(\Sigma)}{2k}\right), \quad (8)$$

*where $\lambda_{rank(\Sigma)}$ denotes the smallest non-zero eigenvalue of $\Sigma$, any estimator $\tilde{\theta}$ derived from $n$ samples satisfies*

$$\inf_{\tilde{\theta}} \sup_{\theta^* \in \Theta_{B,k}} \mathbb{E}\left[\left\|\tilde{\theta} - \theta^*\right\|_\Sigma^2\right] \ge C\frac{\sigma^2}{\zeta} \frac{k \log\left(1 + \frac{rank\,\Sigma}{2k}\right)}{n}, \quad (9)$$

*where $\zeta$ is defined in (7).*

The proof of Theorem 3.1 is provided in Appendix E.1.

**Corollary 3.1.** *Suppose the assumptions in Theorem 3.1 hold. For a nonsingular Gram matrix $\Sigma$, the minimax lower bound is of the order $\Omega\left((k/n) \log(d/k)\right)$.*

*Remark* 3.1. Theorem 3.1 shows that the order of the minimax lower bound depends on rank($\Sigma$), rather than the ambient dimension $d$.

Compared to the non-sparse case which has a lower bound of $\Omega(d)$ (Shah et al., 2016; Zhu et al., 2023), the lower bound in Theorem 3.1 reduces the dimension dependency to $\Omega(k \log(d/k))$.

## 3.2. Upper Bounds

To derive upper bounds on the estimation errors of the estimators under consideration, we assume that $F$ in the sparse RUM (5) is strongly log-concave in a neighborhood of the origin.

**Assumption 3.1** (Strong log-concavity). For the function $F$ in the sparse RUM (5), there exists $\gamma > 0$ such that for any $t \in [-BL/\sigma, BL/\sigma]$,

$$\frac{d^2}{dt^2} \left( -\log F(t) \right) \geq 2\gamma. \tag{10}$$

Assumption 3.1 is satisfied by the BTL (where $F$ is the sigmoid function) and TM models (where $F$ is the cumulative distribution function of the standard Gaussian distribution). This assumption is common in prior works, such as Shah et al. (2016) and Zhu et al. (2023).

Define the parameter $\omega$ as the supremum of the logarithmic derivative of $F$ over the interval $[-BL/\sigma, BL/\sigma]$, i.e.,

$$\omega := \sup_{t \in [-BL/\sigma, BL/\sigma]} \frac{d}{dt} \log F(t). \tag{11}$$

$\omega$ represents the Lipschitz continuity constant of $\log F$ over the given interval. Similar to $B, L, \sigma, \zeta$, we treat $\gamma$ and $\omega$ as fixed. Specifically, when $B = L = 1$, $\gamma = 0.10, \omega = 0.73$ in the BTL model and $\gamma = 0.18, \omega = 1.52$ in the TM model.

### 3.2.1. $\ell_0$-CONSTRAINED ESTIMATOR

Now let us consider the $\ell_0$-constrained maximum likelihood estimator $\hat{\theta}_{\ell_0}^k$, defined as

$$\hat{\theta}_{\ell_0}^k \in \arg\min_{\theta \in \Theta_{B,k}} \mathcal{L}(\theta, \{\xi_i\}_{i=1}^n). \tag{12}$$

Finding such minimizers is computationally intractable in general, as it involves searching over all possible $k$ subset out of $d$-dimensional vector, which takes $\binom{d}{k}$ number of maximum likelihood estimates. Nevertheless, we are interested in its estimation error rate as a theoretical benchmark.

To provide an upper bound on the estimation error of the $\ell_0$-constrained estimator, we begin by introducing some notation. For an index set $S \subset [d] := \{1, 2, \ldots, d\}$ and a vector $x \in \mathbb{R}^d$, we denote $x_S \in \mathbb{R}^{|S|}$ as the vector of $x$ consisting of the elements indexed by $S$, and $|S|$ the cardinality of $S$. We then define the principal submatrix of $\Sigma$ accordingly, i.e.,

$$\Sigma_S := \frac{1}{n} \sum_{i=1}^n (x_{0,i} - x_{1,i})_S (x_{0,i} - x_{1,i})_S^\top \in \mathbb{R}^{|S| \times |S|}. \tag{13}$$

We then make the following assumption.

**Assumption 3.2** (Nonsingularity of submatrices). For each $S$ such that $k \leq |S| \leq 2k$, $\text{rank}(\Sigma_S) = |S|$.

Assumption 3.2 does not impose a full-rank requirement on the Gram matrix $\Sigma$. Moreover, if $\{x_{0,i}, x_{1,i}\}_{i=1}^n$ are randomly sampled from an absolutely continuous probability measure, Assumption 3.2 is satisfied with probability 1.

*Remark* 3.2. Assumption 3.2 is not required if (4) is replaced with the regularized metric

$$\|\hat{\theta} - \theta^*\|_{\Sigma + \lambda I}^2 \tag{14}$$

where $\lambda > 0$ is fixed, and $I$ is the $d \times d$ identity matrix. Adding the regularization term $\lambda I$ to $\Sigma$ ensures that the semi-norm $\|\cdot\|_\Sigma$ becomes a norm $\|\cdot\|_{\Sigma + \lambda I}$. However, adopting (14) as the metric introduces an additive constant term $\lambda B^2$ to the upper bound, similar to Lemma 3.1 in Zhu et al. (2023). We note that this constant term does not affect the order of the bound, as $\lambda$ can be made arbitrarily small. For simplicity, we adopt (4) as the metric in this paper.

**Theorem 3.2** (Minimax optimal rate for the $\ell_0$-constrained estimator). *Consider the sparse RUM (5) with $k \leq d/2$. Suppose Assumption 3.1 and 3.2 hold. With probability at least $1 - \delta$,*

$$\left\| \hat{\theta}_{\ell_0}^k - \theta^* \right\|_\Sigma^2 \leq \frac{24\omega^2\sigma^2}{\gamma^2} \frac{k \log\left(\frac{d}{k}\right) + \log(1/\delta)}{n} \tag{15}$$

*where $\omega$ is defined in (11).*

The proof of Theorem 3.2 is provided in Appendix E.2.

From Theorem 3.1 and 3.2, it follows that the $\ell_0$-constrained estimator defined in (12) achieves minimax optimality over $\Theta_{B,k}$ with respect to $k$ and $n$, and with the full-rank assumption on $\Sigma$, also with respect to $d$.

For sparse preference learning under Assumption 3.1, the sample complexity is of the same order as that of sparse linear regression under sub-Gaussian noise (Vershynin, 2015; Rigollet & Hütter, 2023).

### 3.2.2. $\ell_1$-REGULARIZED ESTIMATOR

One widely adopted approach to overcoming the computational intractability of $\ell_0$-norm constrained problems is to relax the $\ell_0$-norm constraint by incorporating a weighted $\ell_1$-norm term into the objective function, as seen in methods like LASSO (Tibshirani, 1996). Motivated by this approach, this subsection focuses on evaluating the performance of estimating the sparse parameter $\theta^*$ by minimizing the maximum likelihood loss with a regularization term $\beta\|\theta\|_1$. We refer to this $\ell_1$-norm penalized estimator as the $\ell_1$-*regularized estimator*, formally defined as:

$$\hat{\theta}_{\ell_1} \in \arg\min_{\theta \in \Theta_B} \mathcal{L}(\theta, \{\xi_i\}_{i=1}^n) + \beta\|\theta\|_1. \tag{16}$$

Since the $\ell_1$-norm is the convex envelope of the $\ell_0$-norm, the transformed problem becomes convex and can be efficiently

solved using methods such as coordinate descent (Boyd et al., 2011; Peng & Vidal, 2023) and proximal gradient algorithms (Tseng, 2008; Beck & Teboulle, 2009; Becker et al., 2011). Notably, this approach does not require the prior knowledge of $k$.

We define $H$ to characterize the boundedness of the columns of the feature matrix $X$, i.e.,

$$H := \frac{\max_j \|X_j\|_2}{\sqrt{n}} \qquad (17)$$

where $X_j$ denotes the $j$-th column of $X$. We note that the parameter $H$ always exists and satisfies $H \leq L$, as the feature space $\mathcal{D}$ is bounded (see Equation (6)).

**Theorem 3.3** (Slow rate for the $\ell_1$-regularized estimator).
*Consider the sparse RUM (5). Suppose Assumption 3.1 holds. With probability at least $1 - \delta$, the $\ell_1$-regularized estimator (16) with*

$$\beta = \frac{\sqrt{2}\omega H}{\sigma}\sqrt{\frac{\log 2d + \log(1/\delta)}{n}} \qquad (18)$$

*satisfies*

$$\left\|\hat{\theta}_{\ell_1} - \theta^*\right\|_\Sigma^2 \leq \frac{2\sqrt{2}\,\omega H}{\gamma}\,\sigma\,\|\theta^*\|_1 \sqrt{\frac{\log 2d + \log(1/\delta)}{n}}, \qquad (19)$$

*where $\omega$ is defined in (11), and $H$ in (17).*

The proof of Theorem 3.3 is provided in Appendix E.3.

According to (5), we have $\|\theta^*\|_1 \leq B\sqrt{k}$. The estimation error rate in Theorem 3.3 is thus $\mathcal{O}\left(\sqrt{(k/n)\log d}\right)$, which has a gap from the minimax optimal rate $\Theta\left((k/n)\log(d/k)\right)$ in Theorem 3.2. Next, we show that the $\ell_1$-regularized estimator can achieve a *nearly* minimax optimal rate under the following assumption on the spectrum of $\Sigma$.

**Assumption 3.3** (Restricted eigenvalue condition). We assume that the Gram matrix $\Sigma$ satisfies

$$\inf_{1 \leq |S| \leq k}\ \inf_{\theta \in \mathcal{C}_S}\ \frac{\|\theta\|_\Sigma^2}{\|\theta\|_2^2} \geq \frac{1}{2} \qquad (20)$$

where $\mathcal{C}_S := \left\{\theta \in \mathbb{R}^d \mid \theta \neq 0, \|\theta_{S^c}\|_1 \leq 3\|\theta_S\|_1\right\}$.

Assumption 3.3 implies that the smallest eigenvalue of $\Sigma_S$ is lower bounded by a positive constant for all $S$ of cardinality no more than $k$. A stronger version of the assumption requires $\Sigma$ satisfies the *incoherence* condition, namely, $\|\Sigma - I_d\|_{\max} \leq 1/(32k)$, where $\|\cdot\|_{\max}$ denotes the largest absolute value among the elements of a matrix (Bickel et al., 2009; Wainwright, 2019).

**Theorem 3.4** (Fast rate for the $\ell_1$-regularized estimator).
*Consider the sparse RUM (5). Suppose Assumption 3.1 and*

*3.3 hold. With probability at least $1 - \delta$, the $\ell_1$-regularized estimator (16) with*

$$\beta = \frac{4\omega}{\sigma}\sqrt{\frac{\log 2d + \log(1/\delta)}{n}} \qquad (21)$$

*satisfies*

$$\left\|\hat{\theta}_{\ell_1} - \theta^*\right\|_\Sigma^2 \leq \frac{128\omega^2\sigma^2}{\gamma^2}\frac{k\log 2d + \log(1/\delta)}{n} \qquad (22)$$

*and*

$$\left\|\hat{\theta}_{\ell_1} - \theta^*\right\|_2^2 \leq \frac{256\omega^2\sigma^2}{\gamma^2}\frac{k\log 2d + \log(1/\delta)}{n}. \qquad (23)$$

The proof of Theorem 3.4 is provided in Appendix E.4.

Theorem 3.4 establishes that a computationally tractable estimator achieves an estimation error rate of $\mathcal{O}\left((k/n)\log d\right)$ under Assumption 3.3, which is nearly minimax optimal.

*Remark* 3.3. Theorem 3.3 and 3.4 suggest that, to achieve the estimation error rate of the corresponding order, the regularization parameter should scale as $\beta \sim n^{-0.5}$. Yet, the optimal choice of $\beta$ in practice remains unclear. To investigate this, we conduct a hyperparameter search; implementation details are provided in Appendix C.1. As shown in Figure 1, $\beta$ is expected to decrease as the sample size $n$ increases. The red line represents $\log(\beta)$ as a linear function of $\log(n)$ with a slope of $-0.5$. Notably, this line aligns with the valley of the contour map, validating that our theoretical results offer a reasonable guideline for picking $\beta$.

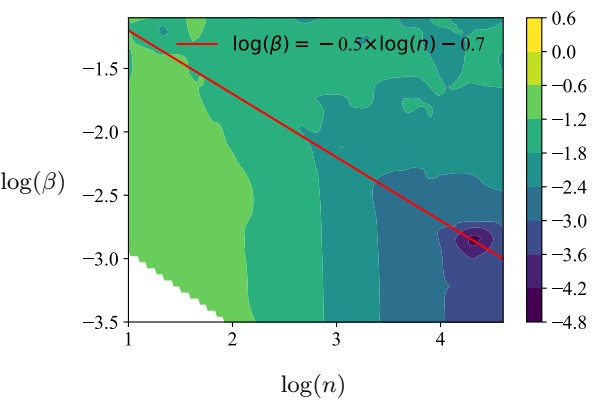

*Figure 1.* Contour of the estimation error of $\|\hat{\theta}_{\ell_1} - \theta^*\|_\Sigma^2$ with respect to $\beta$ and $n$ in the log space for $d = 100, \sigma = 0.1$. The logarithm uses a base of 10.

## 4. Experimental Results

To demonstrate the sample efficiency of the proposed $\ell_1$-regularized estimator, we conduct experiments on synthetic data (Section 4.1) as well as on the task of LLM alignment

(Section 4.2) in two settings: frozen backbone training and full fine-tuning. We defer additional results and discussion to Appendix D. The code can be found at this link[3].

### 4.1. Numerical Evaluation on Synthetic Data

**Experimental setting.** The ground-truth parameter $\theta^*$ is sampled from the set $\{\theta \in \mathbb{R}^{d-1} : \|\theta\|_2 = 1, \|\theta\|_0 = k\}$. Specifically, $k$ out of $d$ coordinates are selected uniformly at random. The value at each selected coordinate is i.i.d. drawn from the standard Gaussian distribution. Finally, the resulting vector is normalized to have a unit Euclidean norm. Each $x_{0,i}$ and $x_{1,i}$ in $\{(x_{0,i}, x_{1,i})\}_{i=1}^n$ are independently sampled from the uniform distribution $\mathcal{U}([0,1]^d)$. The observed preference signal $y_i$ with respect to $x_{0,i}$ and $x_{1,i}$ is generated according to the random utility model, as shown in (1), where $F(t) = \frac{1}{1+\exp(-t)}$ is the sigmoid function. Specifically, $y_i$ is a Bernoulli random variable with parameter $p$ derived from the random utility model. Both the maximum likelihood estimator and the $\ell_1$-regularized estimator are implemented using the SciPy package (Virtanen et al., 2020) with the SLSQP optimization method (Kraft, 1988). To ensure convergence, we set the maximum number of iterations to 1000.

**Results.** Figure 2 compares the $\ell_1$-regularized estimator and the maximum likelihood estimator under varying sparsity ratios $(k/d)$ and sample sizes $(n)$. The estimation error is evaluated using the semi-norm $\|\cdot\|_\Sigma$, defined in (4). In Figure 2a, as the ground-truth parameter $\theta^*$ is increasingly sparse (the ratio $k/d$ decreases), the $\ell_1$-regularized estimator demonstrates superior performance compared to the maximum likelihood estimator, which is agnostic to sparsity level. Similarly, in Figure 2b, the $\ell_1$-regularized estimator consistently exhibits greater sample efficiency, particularly when the sample size $n$ is small. Note that the penalization parameter $\beta$ is selected based on the theoretical results presented in Section 3.2.2 and the outcomes of our hyperparameter search described in Appendix C.1.

### 4.2. Empirical Evaluation

We present proof-of-concept results on real-world datasets to assess the performance of the sparsity-aware methods in reward learning, which is a critical component of RLHF. The performance of a reward model is evaluated based on prediction accuracy on the test dataset. A prediction is considered correct for a given prompt-response pair if the reward model assigns a higher reward to the chosen response than to the rejected response.

**Data.** We train reward models using the `rm-static` dataset (Bai et al., 2022)[4] and `SHP` dataset (Ethayarajh et al., 2022)[5]. `rm-static` is a curated dataset specifically designed for training reward models. The dataset consists of 76.3K samples, each comprising a prompt and a pair of responses, where one response is marked as chosen and the other as rejected, based on annotations provided by human evaluators. An example is shown in Appendix C.2.

**Models and Methods.** We employ the pretrained language models `Pythia-70M` (Biderman et al., 2023) and `Llama-3.2-1B` (Dubey et al., 2024) as the foundation for reward modeling. To adapt such a model, the final layer is replaced with a scalar head to produce reward values for input responses. For the $\ell_1$-regularized method, we add $\beta\|\theta\|_1$ to the original loss function, where $\theta$ represents the parameter of the final layer. As a baseline for comparison, we set $\beta = 0$, namely removing the regularization term. The code is based on `Deepspeed-Chat` (Yao et al., 2023). Detailed parameter settings can be found in Appendix C.3.

**Results for full fine-tuning.** To mitigate the influence of randomness from different random seeds, we fit a quadratic model to capture the relationship between accuracy and the $\ell_1$-norm regularization parameter $\beta$, as shown in Figure 3. The results indicate that applying $\ell_1$ regularization to the last layer leads to a $0.9\%$ improvement in accuracy for both models examined. Furthermore, the $\ell_1$-regularized models (gray curve) consistently outperforms the baseline models (dashed line) across a wide range of $\beta$ values. The empirical results validate the effectiveness of the proposed sparse-aware reward modeling approach, demonstrating its potential value in RLHF.

**Frozen backbone training** In frozen backbone training, only the last layer is trained, while all other parameters (the backbone) remain frozen—a method often referred to as *linear probing* or *feature-based fine-tuning*. Since LLMs have billions of parameters, full fine-tuning can be extremely expensive. By updating only the last layer, memory usage and computation time are drastically reduced. This approach is widely used when computational resources are constrained, data is scarce, or as a baseline before full fine-tuning.

**Results for frozen backbone training** Figure 4 compares the test accuracy of $\ell_1$-regularized reward modeling (orange curve) and the baseline (blue curve). The underlying model is `Llama-3.2-1B`, where the dimensionality of the second-last layer output is $d = 2048$, and the dataset is `rm-static`. Our results show that leveraging spar-

---

[3]https://github.com/yaoyzh/
SparsePreferenceLearning

[4]https://huggingface.co/datasets/Dahoas/
rm-static

[5]https://huggingface.co/datasets/
stanfordnlp/SHP

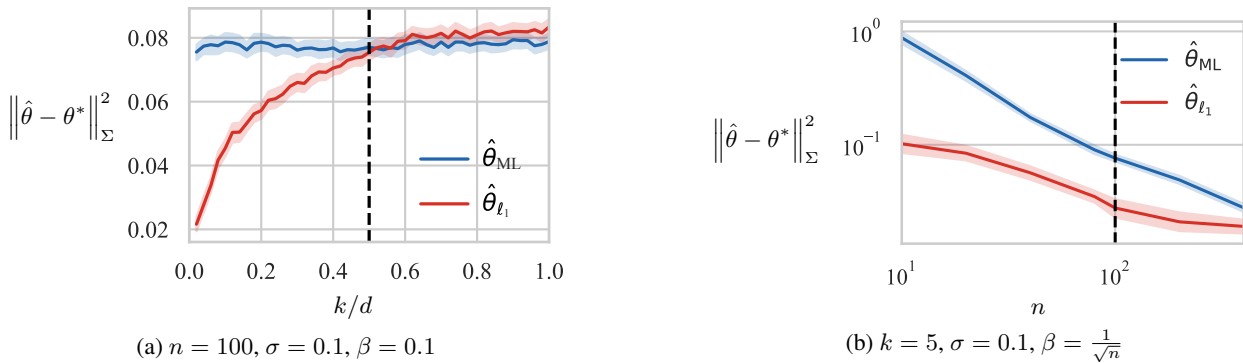

(a) $n = 100$, $\sigma = 0.1$, $\beta = 0.1$

(b) $k = 5$, $\sigma = 0.1$, $\beta = \frac{1}{\sqrt{n}}$

*Figure 2.* Estimation error of $\hat{\theta}_{\ell_1}$ and $\hat{\theta}_{\mathrm{ML}}$. Results are based on 20 repetitions of experiments conducted with dimension $d = 100$.

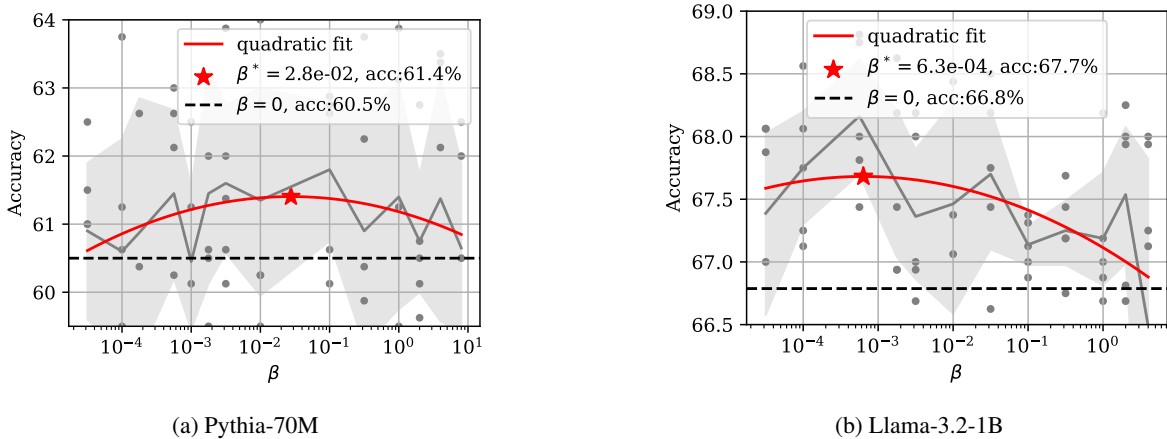

(a) Pythia-70M

(b) Llama-3.2-1B

*Figure 3.* Full fine-tuning: accuracy versus $\ell_1$ regularization parameter $\beta$. Each gray dot represents the accuracy for a specific value of $\beta$ from a single trial. The gray curve in each sub-figure illustrates the average accuracy over five trials (five gray dots) for each specific $\beta$. The red curve represents the quadratic fit across all trials (all gray dots), with the maximum accuracy of the fit curve highlighted. The black dashed line indicates the average accuracy obtained without any regularization. The dataset is `rm-static`.

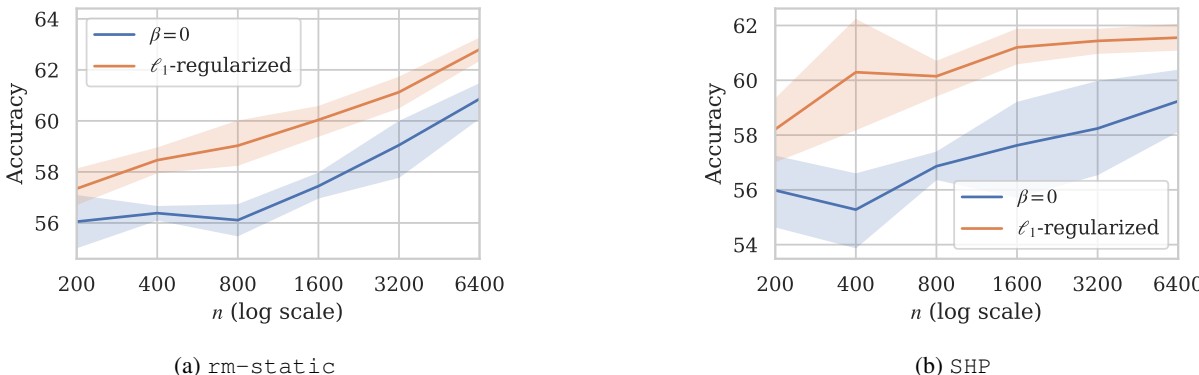

(a) `rm-static`

(b) `SHP`

*Figure 4.* Frozen backbone training: test accuracy vs. $n$. The backbone model is `Llama-3.2-1B`, of which the dimensionality of the second-last layer output is $d = 2048$. We set $\beta = 0.5 \times n^{-0.5}$ for the $\ell_1$-regularization. Each setting is evaluated over 5 trials.

sity can improve accuracy by at least 3%. Notably, we do not manually tune the hyperparameter $\beta$; instead, we set $\beta = 0.5 \times n^{-0.5}$ for the $\ell_1$-regularized method. In both datasets, the $\ell_1$-regularized method (orange curve) consistently outperforms the baseline ($\beta = 0$, blue curve) across all values of $n$. Furthermore, we observe that for

the `rm-static` dataset, the learned parameters has sparsity ratio $k/d \approx 4.5\%$ if $n = 800$ and $k/d \approx 7.5\%$ if $n = 3200$; for the `SHP` dataset: $k/d \approx 4.2\%$ if $n = 800$ and $k/d \approx 7.2\%$ if $n = 3200$. These results show that $\ell_1$ regularization selects a small and informative subset of features, and sparsity regularization is beneficial for reward modeling, leading to higher test accuracy.

## 5. Related Works

Expected utility, originating from mathematical economics, posits that rational agents maximize their utility under uncertainty (Ramsey, 1926; Von Neumann & Morgenstern, 1947). Debreu et al. (1954) established the preference representation theorem, which asserts that any complete, transitive, and continuous preference relation can be represented by a continuous ordinal utility function. This deterministic model is extended to the Random Utility Model (RUM) (McFadden, 1974; 1978; Rosenfeld et al., 2020; Azar et al., 2024; Samuelson, 2024; Sun et al., 2024), incorporating elements such as Gaussian noise (Thurstone, 1994), Gumbel noise (Luce, 1959), and others (Tesauro, 1988; Crammer & Singer, 2003; Chajewska et al., 2001). Furthermore, the RUM can be extended to incorporate multiple utility functions (Moulin, 1985; Eliaz & Ok, 2006; Benson et al., 2018; Pfannschmidt & Hüllermeier, 2020; Benavoli et al., 2023), non-linear utility models based on Gaussian processes (Benavoli & Azzimonti, 2024), and context-dependent models (Seshadri et al., 2019; Bower & Balzano, 2020; Tomlinson & Benson, 2021; 2024).

Apart from utility-based methods, preference learning can be achieved through preference ranking (Haddawy et al., 2003; Fürnkranz & Hüllermeier, 2003; Brazdil et al., 2003; Negahban et al., 2012; Wauthier et al., 2013; Hajek et al., 2014; Rajkumar & Agarwal, 2014; Park et al., 2015; Shah et al., 2016; Seshadri et al., 2020; Chen et al., 2022a) without explicitly modeling utility functions, or by framing it as a classification problem in machine learning contexts (Fürnkranz & Hüllermeier, 2011; Van Cranenburgh et al., 2022). Other closely related fields include ranking learning and ordinal regression, which involve predicting ordered classes for each sample (Frank & Hall, 2001; Kramer et al., 2001; Chu & Keerthi, 2005).

Powerful large language models (LLMs) trained through next-token prediction can generate unhelpful and unsafe content that is misaligned with human instructions (Leike et al., 2018). To address this, a popular approach is to align pretrained models with human instructions through Reinforcement Learning from Human Feedback (RLHF) (Ziegler et al., 2019; Ouyang et al., 2022; Wu et al., 2024). Based on whether an explicit reward function is employed, RLHF methods primarily fall into two categories: reward-based approaches and reward-free approaches. Reward-based RLHF

typically involves two main steps: first, training a reward model based on user feedback to capture human intentions, and second, training a policy using reinforcement learning to optimize the learned reward model. Various enhancements to RLHF have been proposed to improve its efficiency and effectiveness, including accelerated training methods (He et al., 2024) and self-play techniques (Wu et al., 2024). Notably, Direct Preference Optimization (DPO), a reward-free approach, has been demonstrated to yield the same solutions as reward-based RLHF (Rafailov et al., 2024; Xu et al., 2024; Shi et al., 2024), incorporating methods such as reject sampling (Liu et al., 2024). Iterative DPO approaches, as explored by Xiong et al. (2024) and Yuan et al. (2024), leverage the LLM itself as the reward model to provide preference signals, a strategy referred to as self-rewarding. Moreover, Zhu et al. (2023) introduces a pessimistic maximum likelihood estimation (MLE) approach for training policies. The theoretical foundations of these methods often draw from dueling bandit frameworks (Lu et al., 2010; Yue et al., 2012; Saha, 2021; Bengs et al., 2021), with online RLHF approaches initially developed for finite and small state spaces (Xu et al., 2020; Novoseller et al., 2020; Pacchiano et al., 2021) and subsequently generalized to approximate complex functions (Chen et al., 2022b).

Sparse linear models have become a cornerstone of high-dimensional statistics, leveraging the assumption that only a few predictors significantly influence the response. It has been shown that $\ell_1$-based methods, such as LASSO (Tibshirani, 1996) and the Dantzig Selector (Candes & Tao, 2007), can achieve $\mathcal{O}((k/n)\log(d))$ (Candes & Tao, 2006; Bickel et al., 2009) under incoherence conditions, which is close to the minimax rate $\Theta((k/n)\log(d/k))$ (Ye & Zhang, 2010; Rigollet & Tsybakov, 2011; Raskutti et al., 2011; Verzelen, 2012; Candes & Davenport, 2013; Reeves & Gastpar, 2013). Later, Bellec et al. (2018) has shown that the minimax rate is achievable by polynomial time methods. Due to the vast literature on sparsity, we refer readers to Hastie et al. (2015) and Wright & Ma (2022) for comprehensive discussions.

## 6. Conclusion and Future Work

In this paper, we address the challenge of sample-efficient preference learning in high-dimensional settings. Leveraging the sparse random utility model, we establish the minimax optimal rate and propose efficient $\ell_1$-regularized estimators to reduce the sample complexity. Experimental results on synthetic data and LLM alignment datasets validate these theoretical insights, demonstrating that sparsity-aware methods not only reduce sample complexity but also enhance prediction accuracy. For future work, we aim to investigate the optimality of RLHF policies induced by such sparse reward models and further extend the sparsity-induced sample-efficient estimation framework to DPO.

## Acknowledgements

This work was supported in part by the Swiss National Science Foundation under Grant 200364.

## Impact Statement

This paper aims to advance the fields of Preference Learning and LLM Alignment through both theoretical analysis and empirical evaluation of preference learning (reward modeling). The theoretical findings provide insights into the sample complexity of this problem, while the proposed methods demonstrate improved estimation of human ethics and preferences. These advancements enhance the helpfulness and controllability of LLMs, ultimately contributing to societal welfare.

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

# A. Structure of the Appendix

The appendix is structured as follows:

- Appendix B lists all notations used in the main text.

- Appendix C provides detailed experimental settings.

- Appendix D presents additional experimental results for reward modeling with real data in full fine-tuning.

- Appendix E contains proofs for all theorems.

# B. List of Notations

The list of notations applies exclusively to the main text and does not include those used in the proofs in the appendix.

| Symbol | Description |
|---|---|
| $\mathcal{A}$ | Set of alternatives, action space, response space. |
| $a_0, a_1, a_{0,i}, a_{1,i} \in \mathcal{A}$ | Element in $\mathcal{A}$. |
| $\phi : \mathcal{A} \to \mathbb{R}^d$ | Feature map from $\mathcal{A}$ to $\mathbb{R}^d$. |
| $\mathcal{D} \subset \mathbb{R}^d$ | Feature space, $\phi(\mathcal{A})$ or $\phi(s_i, \mathcal{A})$. |
| $x_0, x_1, x_{0,i}, x_{1,i} \in \mathcal{D}$ | Element in $\mathcal{D}$, $x_{0,i} := \phi(a_{0,i})$. |
| $y, y_i \in \{0, 1\}$ | Preference signal indicating the preferred feature vector. |
| $\xi_i := (x_{0,i}, x_{1,i}, y_i)$ | Data sample. |
| $r^* : \mathcal{D} \to \mathbb{R}$ | Ground-truth reward function. |
| $\theta^* \in \mathbb{R}$ | Ground-truth parameter of the reward function $r^*$. |
| $k$ | Number of non-zero elements of $\theta^*$, i.e., $\|\theta^*\|_0$. |
| $d$ | Ambient dimension of the feature space $\mathcal{D}$. |
| $n$ | Number of samples. |
| $P_{Y|(X_0, X_1)}$ | Conditional distribution of $y \in \{0, 1\}$ given $(x_0, x_1)$. |
| $\sigma \in \mathbb{R}_+$ | Randomness level of $y$ or temperature parameter. |
| $F : \mathbb{R} \to [0, 1]$ | Function of the random utility model, and $F(t) = 1 - F(-t)$. |
| $\langle \cdot, \cdot \rangle$ | Euclidean inner product. |
| $\mathcal{L}(\theta; \{\xi_i\}_{i=1}^n),\ \mathcal{L}(\theta)$ | Negative log-likelihood function with respect to dataset $\{\xi_i\}_{i=1}^n$. |
| $\Theta$ | Parameter space. |

| Symbol | Description |
|---|---|
| $B$ | Radius of the parameter space, or norm constraint bound. |
| $\hat{\theta}_{\mathrm{ML}}$ | Maximum likelihood estimator in parameter space $\Theta$. |
| $\hat{\theta},\ \tilde{\theta}$ | Estimate of $\theta^*$. |
| $\hat{r}$ | Estimated reward function associated with $\hat{\theta}$. |
| $X$ | Matrix with the $i$-th row being $(x_{0,i} - x_{1,i})$. |
| $\Sigma$ | Gram matrix, or data covariance matrix, $\Sigma = \frac{1}{n} X^\top X$. |
| $\|\cdot\|_\Sigma$ | Semi-norm induced by positive semi-definite matrix $\Sigma$, i.e., $\|x\|_\Sigma = \sqrt{x^\top \Sigma x}$. |
| $\mathcal{S}$ | Prompt space, or state space. |
| $s_i \in \mathcal{S}$ | Prompt. |
| $\phi(\cdot,\cdot) : \mathcal{S} \times \mathcal{A} \to \mathbb{R}^d$ | Feature mapping, or feature embedding of a given prompt-response pair $(s,a)$. |
| $\Theta_B$ | Parameter space with radius $B$. |
| $\Theta_{B,k}$ | $\Theta_{B,k} := \{\theta \in \mathbb{R}^d : \|\theta\|_2 \le B,\ \|\theta\|_0 \le k\}..$ |
| $L$ | Diameter of the feature space $\mathcal{D}$. |
| $\zeta$ | See definition (7). |
| $\lambda_{\mathrm{rank}(\Sigma)}$ | Smallest non-zero eigenvalue of $\Sigma$. |
| $\gamma$ | See definition (3.1). |
| $\omega$ | See definition (11). |
| $\hat{\theta}_{\ell_0}^k$ | Maximum likelihood estimator in parameter space $\Theta_{B,k}$. |
| $[d]$ | $[d] := \{1,\dots,d\}$. |
| $S \subset [d]$ | Index set. |
| $S^c \subset [d]$ | The set complement of $S$. |
| $|S|$ | Cardinality of set $S$. |
| $x_S \in \mathbb{R}^{|S|}$ | Vector of elements in $x$ indexed by $S$. |
| $\Sigma_S$ | Principal submatrix of $\Sigma$ with respect to $S$; see definition (13). |
| $I,\ I_d \in \mathbb{R}^{d \times d}$ | Identity matrix. |
| $\hat{\theta}_{\ell_1}$ | $\ell_1$-regularized estimator in parameter space $\Theta_B$; see definition (16). |

| Symbol | Description |
|---|---|
| $\| \cdot \|_{\max}$ | The largest absolute value among the elements of a matrix. |
| $H$ | See definition (17). |

## C. Experimental Details

### C.1. Hyperparameter Search

We use `hyperopt` package (Bergstra et al., 2013) to conduct a hyperparameter search for $\beta$ with $n$ varying from 10 to $10^5$. Figure 1 is a contour of estimation errors $\|\hat{\theta}_{\ell_1} - \theta^*\|_{\Sigma}^2$ over a series of experiments with different choices of $n$, $\beta$, and random seeds, extrapolated through cubic polynomials. The red line in the figure represents $\log(\beta)$ as a linear function of $\log(n)$ with slope $-0.5$. Our results show that $\beta$ given by the theoretical results $\beta \sim n^{-0.5}$ in Theorem 3.3 and 3.4 does lead to the valley in the contour surface. All experiments are repeated for 20 times.

### C.2. Illustrative Examples for Datasets

> **An illustrative example for a data sample in `rm-static`**
>
> **Prompt:**
> Human: Can you describe the steps to clean fingerprints and smudges from a laptop screen Assistant: Yes, certainly. To clean your screen, you first need to use a microfiber cloth or soft, damp cloth to gently wipe down the surface of the screen. Next, you'll want to grab a soft, lint-free, microfiber cleaning cloth and gently rub it back and forth across the screen to remove fingerprints and smudges. Human: Can I spray isopropyl alcohol onto the cloth and clean it that way? Assistant:
>
> **Chosen response:**
> Yes, you can do that to help the cloth pick up even more dirt from the screen. Be sure to always use a clean, soft cloth, not a piece of scratchy, roughened, or textured material, and make sure it's lint-free.
>
> **Rejected response:**
> Yes, you can spray it directly onto the cloth.

In addition to `rm-static` mentioned in Section 4.2, we also use the dataset `SHP` (Ethayarajh et al., 2022)[6]. As noted by the authors of Ethayarajh et al. (2022):

> Most notably, all the data in `SHP` is naturally occurring and human-written, whereas the responses in `rm-static` are machine-written, giving us two very different distributions that can complement each other.

### C.3. Parameter Setting for Real-Data Experiments

The learning rate is set to $10^{-5}$, and the weight decay is set to 0.1. The batch size is 8 for `Pythia-70M`, 16 for `Llama-3.2-1B` and 32 for `Llama-3.2-3B`, and the training runs for 1 epoch. The regularization hyperparameter $\beta$ for the $\ell_1$-regularized method is selected from the range $10^{[-4.5:0.5:0]} \cup \{2, 4, 8\}$. Each $\beta$ value, including $\beta = 0$, is evaluated across 5 trials with random seeds in $\{0, 1, 2, 3, 4\}$ for `Pythia-70M` and `Llama-3.2-1B`, and 3 trials with random seeds in $\{0, 1, 2\}$ for `Llama-3.2-3B`.

## D. Additional Experimental Results for Reward Modeling

### D.1. Full Fine-Tuning

This section presents additional results for full fine-tuning, where all backbone model parameters are fine-tuned, consistent with Section 4.2.

---

[6] `https://huggingface.co/datasets/stanfordnlp/SHP`

We conduct experiments using three base models: `Pythia-70M`, `Llama-3.2-1B`, and `Llama-3.2-3B`. The results are summarized in Table 3 for `rm-static` and in Table 4 for `SHP`. Each reported value represents the average over five trials for `Pythia-70M` and `Llama-3.2-1B`, and three trials for `Llama-3.2-3B`. For the $\ell_1$-regularized estimator, the displayed value corresponds to the regularization parameter that achieved the highest average accuracy. From the tables, we observe that: 1) larger models consistently produce more accurate predictions, which is expected, and 2) the $\ell_1$-regularized models consistently outperforms baseline models, demonstrating its sample efficiency.

*Table 3.* Test accuracy on `rm-static`

| Model | Baseline (%) | $\ell_1$-Regularized (%) |
|---|---|---|
| Pythia-70M | 60.5 | 61.8 |
| Llama-3.2-1B | 66.8 | 67.7 |
| Llama-3.2-3B | 68.4 | 69.4 |

*Table 4.* Test accuracy on `SHP`

| Model | Baseline (%) | $\ell_1$-Regularized (%) |
|---|---|---|
| Pythia-70M | 60.8 | 62.1 |
| Llama-3.2-1B | 64.4 | 65.5 |
| Llama-3.2-3B | 66.5 | 67.3 |

# E. Proofs

## E.1. Proof of Theorem 3.1

Before proceeding, we first prepare some ingredients.

**Lemma E.1** (Sparse Varshamov-Gilbert, Lemma 4.14 in Rigollet & Hütter (2023))**.** *For any two integers $k$ and $d$ such that $1 \leq k \leq d/8$ and Hamming distance $\mathrm{Ham}(\cdot, \cdot)$, there exist binary vectors $w_1, \ldots, w_M \in \{0,1\}^d$ such that*

1. *$\mathrm{Ham}(w_i, w_j) \geq \frac{k}{2}$ for all $i \neq j$;*

2. *$\log(M) \geq \frac{k}{8} \log\left(1 + \frac{d}{2k}\right)$;*

3. *$\|w_j\|_0 = k$ for all $j \in [M]$.*

**Lemma E.2** (Upper bound for pairwise KL divergence)**.** *For any pair of $\theta_1, \theta_2 \in \Theta_B := \{\theta \in \mathbb{R}^d : \|\theta\|_2 \leq B\}$, we have*

$$D_{KL}\left(P_{\theta_1}\left(\{\xi_i\}_{i=1}^n\right) \| P_{\theta_2}\left(\{\xi_i\}_{i=1}^n\right)\right) \leq \frac{n\zeta}{\sigma^2} \|\theta_1 - \theta_2\|_\Sigma^2$$

*where $P_{\theta_j}\left(\{\xi_i\}_{i=1}^n\right) = \Pi_{i=1}^n F\left(\frac{\langle\theta_j, x_{0,i} - x_{1,i}\rangle}{\sigma}\right)^{(1-y_i)} \left(1 - F\left(\frac{\langle\theta_j, x_{0,i} - x_{1,i}\rangle}{\sigma}\right)\right)^{y_i}$ is the joint distribution of $Y_1, \ldots, Y_n$ given $\{(x_{0,i}, x_{1,i})\}_{i=1}^n$ with parameter $\theta_j$.*

See Appendix E.5 for the proof of Lemma E.2.

**Lemma E.3** (Pairwise Fano minimax lower bound, The local Fano method in Duchi (2024)). *We call a subset of vectors* $\{\theta_1, \ldots, \theta_M\} \subset \Theta$ *a* $(\nu, \eta)$-packing of $\Theta$ in a pseudo-metric $\rho$ if

$$\min_{\substack{i,j \in [M] \\ i \neq j}} \rho\left(\theta_i - \theta_j\right) \geq \nu, \quad \text{and} \quad \frac{1}{\binom{M}{2}} \sum_{\substack{i,j \in [M] \\ i \neq j}} D_{KL}\left(P(\theta_i) \parallel P(\theta_j)\right) \leq \eta.$$

*If we can construct a* $(\nu, \eta)$-packing with cardinality $M$, then the minimax risk in the square of the pseudo-metric $\rho$ is lower bounded as

$$\inf_{\tilde{\theta}} \sup_{\theta^* \in \Theta} \mathbb{E}\left[\rho\left(\tilde{\theta} - \theta^*\right)^2\right] \geq \frac{\nu^2}{2}\left(1 - \frac{\eta + \log 2}{\log M}\right) \tag{24}$$

With the above three lemmas at hand, we can now cook up the lower bound in Theorem 3.1 as follows. We first apply Lemma E.1 by replacing $d$ in Lemma E.1 with $\text{rank}(\Sigma)$ to obtain a subset of binary vectors $\{v_1, \ldots, v_M\} \in \{0,1\}^{\text{rank}(\Sigma)}$ as such. Then for each $j \in [M]$, append $(d - \text{rank}(\Sigma))$ zeros to the bottom of $v_j$ and get a $d$-dimensional binary vector $w_j$, i.e.,

$$w_j = [v_j^\top \ 0 \ \ldots \ 0] \in \{0,1\}^d \tag{25}$$

Since $\Sigma$ is symmetric and positive semi-definite, it has an orthogonal diagonalization $\Sigma = U^\top \Lambda U$, where $U \in \mathbb{R}^{d \times d}$ is an orthogonal matrix, and $\Lambda$ is a diagonal matrix with non-negative elements in descending order. Let diagonal matrix $\Lambda^\dagger$ denote the Moore-Penrose pseudo-inverse of $\Lambda$. Let $\theta_1, \ldots, \theta_M$ be such that for each $j \in [M]$,

$$\theta_j = \frac{\sigma}{8\sqrt{\zeta}} \sqrt{\frac{\log\left(1 + \frac{\text{rank}(\Sigma)}{2k}\right)}{n}} \, U^\top \sqrt{\Lambda^\dagger} \, w_j$$

Then,

$$\begin{aligned}
\|\theta_j\|_2 &= \frac{\sigma}{8\sqrt{\zeta}} \sqrt{\frac{\log\left(1 + \frac{\text{rank}(\Sigma)}{2k}\right)}{n}} \left\| U^\top \sqrt{\Lambda^\dagger} \, w_j \right\|_2 \\
&= \frac{\sigma}{8\sqrt{\zeta}} \sqrt{\frac{\log\left(1 + \frac{\text{rank}(\Sigma)}{2k}\right)}{n}} \left\| \sqrt{\Lambda^\dagger} \, w_j \right\|_2 \\
&\leq \frac{\sigma}{8\sqrt{\zeta}} \sqrt{\frac{k \log\left(1 + \frac{\text{rank}(\Sigma)}{2k}\right)}{n}} \max\left(\text{diag}(\sqrt{\Lambda^\dagger})\right) \leq B
\end{aligned}$$

The last inequality holds as we assume

$$n \geq \frac{\sigma^2}{64 B^2 \zeta \lambda_{\text{rank}(\Sigma)}} k \log\left(1 + \frac{\text{rank}(\Sigma)}{2k}\right) = \frac{\sigma^2 \max\left(\text{diag}(\Lambda^\dagger)\right)}{64 B^2 \zeta} k \log\left(1 + \frac{\text{rank}(\Sigma)}{2k}\right)$$

Furthermore, we have

$$\begin{aligned}
\|\theta_i - \theta_j\|_\Sigma^2 &= (\theta_i - \theta_j)^\top \Sigma (\theta_i - \theta_j) \\
&= \frac{\sigma^2}{64\zeta} \frac{\log\left(1 + \frac{\text{rank}(\Sigma)}{2k}\right)}{n} (w_i - w_j)^\top \sqrt{\Lambda^\dagger} U U^\top \Lambda U U^\top \sqrt{\Lambda^\dagger} (w_i - w_j) \\
&= \frac{\sigma^2}{64\zeta} \frac{\log\left(1 + \frac{\text{rank}(\Sigma)}{2k}\right)}{n} (w_i - w_j)^\top \sqrt{\Lambda^\dagger} \Lambda \sqrt{\Lambda^\dagger} (w_i - w_j) \\
&= \frac{\sigma^2}{64\zeta} \frac{\log\left(1 + \frac{\text{rank}(\Sigma)}{2k}\right)}{n} \|w_i - w_j\|_2^2
\end{aligned}$$

where the last equality holds because the last $d - \text{rank}(\Sigma)$ entries of $w_i, w_j$ are zeros according to the way we construct $w$'s (c.f. (25)), and each of the first $\text{rank}(\Sigma)$ diagonal elements in $\Lambda$ is non-zero. Furthermore, by Lemma E.1, each element of $(w_i - w_j)$ is in $\{-1, 0, 1\}$, and hence the Hamming distance between $w_i$ and $w_j$ can be bounded as

$$\frac{k}{2} \le \text{Ham}(w_i, w_j) = \|w_i - w_j\|_2^2 = \|w_i - w_j\|_0 \le \|w_i\|_0 + \|w_j\|_0 = 2k$$

Then,

$$\frac{\sigma}{8\sqrt{\zeta}} \sqrt{\frac{k \log\left(1 + \frac{d}{2k}\right)}{2n}} \le \|\theta_i - \theta_j\|_\Sigma \le \frac{\sigma}{8\sqrt{\zeta}} \sqrt{\frac{2k \log\left(1 + \frac{d}{2k}\right)}{n}}$$

By Lemma E.2, we have constructed a $(\nu, \eta)$-packing of cardinality $M$ with

$$\nu = \frac{\sigma}{8\sqrt{\zeta}} \sqrt{\frac{k \log\left(1 + \frac{\text{rank}(\Sigma)}{2k}\right)}{2n}}, \quad \eta = \frac{k \log\left(1 + \frac{\text{rank}(\Sigma)}{2k}\right)}{32}, \quad \log M \ge \frac{k}{8} \log\left(1 + \frac{\text{rank}(\Sigma)}{2k}\right)$$

Apply Lemma E.3, and we get

$$\inf_{\tilde{\theta}} \sup_{\theta^* \in \Theta_{B,k}} \mathbb{E}\left[\left\|\tilde{\theta} - \theta^*\right\|_\Sigma^2\right] \ge \frac{\nu^2}{2}\left(1 - \frac{\eta + \log 2}{\log M}\right)$$

$$= \frac{1}{128\zeta}\sigma^2 \frac{k \log\left(1 + \frac{\text{rank}(\Sigma)}{2k}\right)}{2n}\left(\frac{3}{4} - \frac{8\log 2}{k \log\left(1 + \frac{\text{rank}(\Sigma)}{2k}\right)}\right)$$

$$> \frac{1}{128\zeta}\sigma^2 \frac{k \log\left(1 + \frac{\text{rank}(\Sigma)}{2k}\right)}{2n}\frac{1}{4}$$

$$= \frac{\sigma^2}{1024\zeta} \frac{k \log\left(1 + \frac{\text{rank}(\Sigma)}{2k}\right)}{n}$$

where the last inequality holds if $k \ge 7$. To see it, notice that as $1 \le k \le \frac{\text{rank}(\Sigma)}{8}$, it holds that $\log(1 + \frac{\text{rank}(\Sigma)}{2k}) \ge \log(5)$. If $k \ge 7$, then $k \log(1 + \frac{\text{rank}(\Sigma)}{2k}) \ge 7 \log(5) > 16 \log(2)$, and hence $\frac{8 \log 2}{k \log\left(1 + \frac{\text{rank}(\Sigma)}{2k}\right)} > \frac{1}{2}$.

For $1 \le k \le 6$, let $\mathcal{W} \subset \mathbb{R}^d$ be a set of vectors of cardinality $2^{\text{rank}(\Sigma)}$ such that for $w_j \in \mathcal{W}$, $\|w_j\|_0 = 1$, $\mathbb{1}^\top w_j \in \{\pm 1\}$, and the last $(d - \text{rank}(\Sigma))$ elements of $w_j$ are all zeros. This means $w_j \in \mathcal{W}$ only has one non-zero element, this element is either $1$ or $-1$, and this non-zero element can only appear in the first $\text{rank}(\Sigma)$ entries. In this way, for any pair $w_i, w_j \in \mathcal{W}$ such that $w_i \ne w_j$, it holds that $2 \le \|w_i - w_j\|_2^2 \le 4$. We construct a $(\nu', \eta')$-packing by letting $\theta_j = \frac{\sigma}{8\sqrt{\zeta}} \sqrt{\frac{\log\left(1 + \frac{\text{rank}(\Sigma)}{2k}\right)}{n}} U^\top \sqrt{\Lambda^\dagger} w_j$. Then, for the same reason as above, we have $\|\theta_j\|_2 \le B$ by our assumption on $n$. Also,

$$\frac{\sigma}{8\sqrt{\zeta}} \sqrt{\frac{2 \log(1 + \text{rank}(\Sigma)/2k)}{n}} \le \|\theta_i - \theta_j\|_\Sigma \le \frac{\sigma}{8\sqrt{\zeta}} \sqrt{\frac{4 \log(1 + \text{rank}(\Sigma)/2k)}{n}}$$

Hence, $\nu' = \frac{\sigma}{8\sqrt{\zeta}} \sqrt{\frac{2 \log(1 + \text{rank}(\Sigma)/2k)}{n}}$, $\eta' = \frac{1}{16} \log(1 + \text{rank}(\Sigma)/2k)$, $\log M' = \text{rank}(\Sigma) \log 2$. Apply Lemma E.3, and we get a lower bound

$$\frac{\sigma^2 \log(1 + \text{rank}(\Sigma)/2k)}{64\zeta n}\left(\frac{7}{8} - \frac{\log(1 + \text{rank}(\Sigma)/2k)}{\text{rank}(\Sigma)16 \log 2}\right) \ge \frac{3\sigma^2}{256\zeta} \frac{\log(1 + \text{rank}(\Sigma)/2k)}{n}$$

because $\frac{\log(1 + \text{rank}(\Sigma)/2k)}{\text{rank}(\Sigma)16 \log 2} \le \frac{1}{2}$. We can rewrite this lower bound as

$$\frac{C\sigma^2}{\zeta} \frac{k \log(1 + \text{rank}(\Sigma)/2k)}{n}$$

as $k$ is $\Theta(1)$, and this lower bound is of the same order as the one for $k \ge 7$.

### E.2. Proof of Theorem 3.2

For simplicity, we denote $\mathcal{L}(\theta) := \mathcal{L}(\theta; \{\xi_i\}_{i=1}^n)$. Before proceeding, we define $\ell_S(\theta; \{\xi_i\}_{i=1}^n)$ for an index set $S \subset [d]$ as

$$\ell_S(\theta) = \ell_S(\theta; \{\xi_i\}_{i=1}^n) := -\frac{1}{n} \sum_{i=1}^n \log \left( \mathbb{1}_{\{y_i=0\}} \cdot F\left( \frac{\langle \theta_S, (x_{0,i} - x_{1,i})_S \rangle}{\sigma} \right) \right.$$
$$\left. + \mathbb{1}_{\{y_i=1\}} \cdot F\left( \frac{-\langle \theta_S, (x_{0,i} - x_{1,i})_S \rangle}{\sigma} \right) \right)$$

Then, for any $\theta$ such that $\text{supp}(\theta) \subset S$, it holds that $\|\theta\|_\Sigma = \|\theta_S\|_{\Sigma_S}$, $\mathcal{L}(\theta) = \ell_S(\theta)$, and $(\nabla \ell_S(\theta))_S = (\nabla \mathcal{L}(\theta))_S$. For $\theta^*$, we define $\mathcal{S} := \{S \subset [d] : |S| \le 2k, \ \text{supp}(\theta^*) \subset S\}$.

The following two lemmas play a role in the upper bounds in this paper. More specific versions of these two lemmas are used for upper bounding the estimation error of the maximum likelihood estimator with discrete $\mathcal{D}$ in Shah et al. (2016). For completeness, we put the proof of Lemma E.4 in Appendix E.6 and the proof of Lemma E.5 in Appendix E.7.

**Lemma E.4.** *If $F$ satisfies the strong log-concavity assumption (10), then for any non-empty index set $S \subset [d]$,*

$$\ell_S(\theta^* + \theta') - \ell_S(\theta^*) - \langle \nabla \ell_S(\theta^*), \theta' \rangle \ge \frac{\gamma}{\sigma^2} \|\theta'_S\|_{\Sigma_S}^2 \quad \forall \theta'_S \in \mathbb{R}^d \text{ such that } \theta^* + \theta' \in \Theta_B$$

**Lemma E.5.** *For any non-empty index set $S \subset [d]$, $(\nabla \ell_S(\theta^*))_S = -\frac{1}{n\sigma} X_S^\top V_S$, where*

$$X_S := [(x_{0,1} - x_{1,1})_S, \cdots, (x_{0,n} - x_{1,n})_S]^\top \in \mathbb{R}^{n \times |S|}$$

*and $V_S \in \mathbb{R}^n$ is a random vector with independent components such that $\mathbb{E}[V_S] = 0$, and $\|V_S\|_\infty \le \zeta$.*

By the definition of the $\ell_0$-constrained estimator,

$$\mathcal{L}(\hat{\theta}_{\ell_0}^k) = \min_{\theta \in \Theta_{B,k}} \mathcal{L}(\theta) \le \mathcal{L}(\theta^*)$$

Let $\hat{S} := \text{supp}(\hat{\theta}_{\ell_0}^k) \cup \text{supp}(\theta^*)$. Then, $|\hat{S}| \le 2k$, and $\hat{S} \in \mathcal{S}$. Notice that $\hat{S}$ is a function of $\{\xi_i\}_{i=1}^n$, and hence a random variable. Moreover,

$$\ell_{\hat{S}}\left( \hat{\theta}_{\ell_0}^k \right) \le \ell_{\hat{S}}(\theta^*), \quad \text{and} \quad \left( \nabla \ell_{\hat{S}}(\theta^*) \right)_{\hat{S}^c} = 0$$

By Lemma E.4, since $F$ satisfies (10), we have

$$\frac{\gamma}{\sigma^2} \left\| \left( \hat{\theta}_{\ell_0}^k - \theta^* \right)_{\hat{S}} \right\|_{\Sigma_{\hat{S}}}^2 \le \left| \left\langle \nabla \ell_{\hat{S}}(\theta^*), \hat{\theta}_{\ell_0}^k - \theta^* \right\rangle \right|$$
$$= \left| \left\langle (\nabla \ell_{\hat{S}}(\theta^*))_{\hat{S}}, \left( \hat{\theta}_{\ell_0}^k - \theta^* \right)_{\hat{S}} \right\rangle \right|$$
$$\le \left\| \left( \hat{\theta}_{\ell_0}^k - \theta^* \right)_{\hat{S}} \right\|_{\Sigma_{\hat{S}}} \left\| (\nabla \ell_{\hat{S}}(\theta^*))_{\hat{S}} \right\|_{\Sigma_{\hat{S}}^{-1}}$$

where the last inequality is by the Cauchy-Schwarz inequality for dual norms. Hence,

$$\left\| \left( \hat{\theta}_{\ell_0}^k - \theta^* \right)_{\hat{S}} \right\|_{\Sigma_{\hat{S}}}^2 \le \frac{\sigma^4}{\gamma^2} \left\| (\nabla \ell_{\hat{S}}(\theta^*))_{\hat{S}} \right\|_{\Sigma_{\hat{S}}^{-1}}^2$$

We now need to upper bound $\left\| (\nabla \ell_{\hat{S}}(\theta^*))_{\hat{S}} \right\|_{\Sigma_{\hat{S}}^{-1}}^2$. By Lemma E.5, $\left( \nabla \ell_{\hat{S}}(\theta^*) \right)_{\hat{S}} = -\frac{1}{n\sigma} X_{\hat{S}}^\top V_{\hat{S}}$, where

$$X_{\hat{S}} := \left[ (x_{0,1} - x_{1,1})_{\hat{S}}, \cdots, (x_{0,n} - x_{1,n})_{\hat{S}} \right]^\top \in \mathbb{R}^{n \times |\hat{S}|}$$

and $V_{\hat{S}} \in \mathbb{R}^n$ is a random vector with independent components defined as

$$(V_{\hat{S}})_i := \begin{cases} \frac{F'(\langle \theta_{\hat{S}}, (x_{0,i} - x_{1,i})_{\hat{S}} \rangle / \sigma)}{F(\langle \theta_{\hat{S}}, (x_{0,i} - x_{1,i})_{\hat{S}} \rangle / \sigma)}, & \text{w.p. } F\left( \langle \theta_{\hat{S}}, (x_{0,i} - x_{1,i})_{\hat{S}} \rangle / \sigma \right) \\ \frac{-F'(\langle \theta_{\hat{S}}, (x_{0,i} - x_{1,i})_{\hat{S}} \rangle / \sigma)}{1 - F(\langle \theta_{\hat{S}}, (x_{0,i} - x_{1,i})_{\hat{S}} \rangle / \sigma)}, & \text{w.p. } 1 - F\left( \langle \theta_{\hat{S}}, (x_{0,i} - x_{1,i})_{\hat{S}} \rangle / \sigma \right) \end{cases}$$

Then, $\mathbb{E}[(V_{\hat{S}})_i] = 0$ and $|(V_{\hat{S}})_i| \le \omega$ by the definition (11). Define

$$M_{\hat{S}} := \frac{\sigma^2}{\gamma^2 n^2} X_{\hat{S}} \Sigma_{\hat{S}}^{-1} X_{\hat{S}}^\top \in \mathbb{R}^{d \times d}$$

Then, $\left\| (\nabla \ell_{\hat{S}}(\theta^*))_{\hat{S}} \right\|_{\Sigma_S^{-1}}^2 = \frac{1}{n^2 \sigma^2} V_{\hat{S}}^\top X_{\hat{S}} \Sigma_{\hat{S}}^{-1} X_{\hat{S}}^\top V_{\hat{S}} = \frac{1}{n^2 \sigma^2} \frac{\gamma^2 n^2}{\sigma^2} V_{\hat{S}}^\top M_{\hat{S}} V_{\hat{S}} = \frac{\gamma^2}{\sigma^4} V_{\hat{S}}^\top M_{\hat{S}} V_{\hat{S}}$, and

$$\left\| \left( \hat{\theta}_{\ell_0}^k - \theta^* \right)_{\hat{S}} \right\|_{\Sigma_{\hat{S}}}^2 \le V_{\hat{S}}^\top M_{\hat{S}} V_{\hat{S}}$$

Since $\Sigma_{\hat{S}} \in \mathbb{R}^{|\hat{S}| \times |\hat{S}|}$ is positive definite by assumption, it has an orthogonal diagonalization $\Sigma_{\hat{S}} = U_{\hat{S}}^\top \Lambda_{\hat{S}} U_{\hat{S}} = \frac{1}{n} X_{\hat{S}}^\top X_{\hat{S}}$, where $U_{\hat{S}} \in \mathbb{R}^{|\hat{S}| \times |\hat{S}|}$ is an orthogonal matrix, and $\Lambda_{\hat{S}}$ is a diagonal matrix with positive diagonal elements. Then, $X_{\hat{S}} = \sqrt{n} \Lambda_{\hat{S}}^{1/2} U_{\hat{S}}$, and

$$M_{\hat{S}} = \frac{\sigma^2}{\gamma^2 n} \Lambda_{\hat{S}}^{1/2} U_{\hat{S}} \left( U_{\hat{S}}^\top \Lambda_{\hat{S}} U_{\hat{S}} \right)^{-1} U_{\hat{S}}^\top \Lambda_{\hat{S}}^{1/2} = \frac{\sigma^2}{\gamma^2 n} I_{|\hat{S}|}$$

Therefore,

$$\mathrm{tr}(M_{\hat{S}}) = \frac{|\hat{S}| \sigma^2}{\gamma^2 n} \le \frac{2k\sigma^2}{n\gamma^2}, \quad \mathrm{tr}(M_{\hat{S}}^2) = \frac{|\hat{S}| \sigma^4}{\gamma^4 n^2} \le \frac{2k\sigma^4}{n^2 \gamma^4}, \quad \|M_{\hat{S}}\|_2 = \lambda_{\max}(M_{\hat{S}}) = \frac{\sigma^2}{\gamma^2 n}$$

We then apply Lemma E.6 to get a high probability upper bound for $V_{\hat{S}}^\top M_{\hat{S}} V_{\hat{S}}$.

**Lemma E.6** (Bernstein's inequality for sub-Gaussian in quadratic form, Theorem 2.1 in Hsu et al. (2012)). *Let $A \in \mathbb{R}^{d \times d}$ be a matrix, and $\Sigma = A^\top A$. Suppose that $x = (x_1, \ldots, x_n)$ is a random vector such that, for some $\mu \in \mathbb{R}^n$ and $\sigma \ge 0$, it holds for all $\alpha \in \mathbb{R}^n$ that $\mathbb{E}[\exp(\alpha^\top (x - \mu))] \le \exp(\|\alpha\|_2^2 \sigma^2 / 2)$. For any $t > 0$,*

$$\mathbb{P}\left[ \|Ax\|^2 > \sigma^2 \left( \mathrm{tr}(\Sigma) + 2\sqrt{\mathrm{tr}(\Sigma^2)t} + 2\|\Sigma\|_2 t \right) + \mathrm{tr}(\Sigma \mu \mu^\top) \left( 1 + 2 \left( \frac{\|\Sigma\|_2^2}{\mathrm{tr}(\Sigma^2)} t \right)^{\frac{1}{2}} \right) \right] \le e^{-t}$$

Since $V_{\hat{S}}$ is sub-Gaussian with parameter $\omega^2$, by Lemma E.6,

$$\mathbb{P}\left[ V_{\hat{S}}^\top M_{\hat{S}} V_{\hat{S}} > \omega^2 \left( \mathrm{tr}(M_{\hat{S}}) + 2\sqrt{\mathrm{tr}(M_{\hat{S}}^2)t} + 2\|M_{\hat{S}}\|_2 t \right) \right] \le e^{-t}$$

$$\implies \mathbb{P}\left[ V_{\hat{S}}^\top M_{\hat{S}} V_{\hat{S}} > \omega^2 \left( \frac{2k\sigma^2}{n\gamma^2} + 2\sqrt{\frac{2k\sigma^4}{n^2\gamma^4}t} + 2\frac{\sigma^2}{\gamma^2 n} t \right) \right] \le e^{-t}$$

Equivalently, with probability at least $1 - \delta'$,

$$\left\| \left( \hat{\theta}_{\ell_0}^k - \theta^* \right)_{\hat{S}} \right\|_{\Sigma_{\hat{S}}}^2 \le V_{\hat{S}}^\top M_{\hat{S}} V_{\hat{S}} \le \frac{2\omega^2 \sigma^2}{\gamma^2} \frac{\left( \sqrt{k} + \sqrt{\log(1/\delta')} \right)^2}{n} := t'$$

We notice that for any deterministic $S \in \mathcal{S}$ and $\hat{\theta} \in \Theta_{B,k}$ such that $S = \mathrm{supp}(\hat{\theta}) \cup \mathrm{supp}(\theta^*)$ and $\mathcal{L}(\hat{\theta}) \le \mathcal{L}(\theta^*)$, the above reasoning holds by replacing $\hat{S}$ and $\hat{\theta}_{\ell_0}^k$ with $S$ and $\hat{\theta}$, respectively. In other words, for any index set $S_k$ such that $|S_k| \le k$, for any $\hat{\theta} \in \Theta_{B,k}$ such that $S_k = \mathrm{supp}(\hat{\theta})$ and $\mathcal{L}(\hat{\theta}) \le \mathcal{L}(\theta^*)$, let $S = S_k \cup \mathrm{supp}(\theta^*) \in \mathcal{S}$, and then

$$\mathbb{P}\left[ \left\| \left( \hat{\theta} - \theta^* \right)_S \right\|_{\Sigma_S}^2 \ge t' \right] \le \delta'$$

The cardinality of $\mathcal{S}$ is $\sum_{i=0}^k \binom{d-k}{i}$, and $\sum_{i=0}^k \binom{d-k}{i} \le \left( \frac{de}{k} \right)^k$ because $k \le d/2$. Since $\hat{S} \in \mathcal{S}$ is a random variable, we

apply the union bound for all possible $\hat{S}$, and get

$$
\begin{aligned}
\mathbb{P}\left[\left\|\left(\hat{\theta}_{\ell_0}^k - \theta^*\right)_{\hat{S}}\right\|_{\Sigma_{\hat{S}}}^2 \geq t'\right] &\leq \mathbb{P}\left[\max_{S \in \mathcal{S}}\left\|\hat{\theta}_S - \theta_S^*\right\|_{\Sigma_S}^2 \geq t'\right] \\
&\leq \sum_{S \in \mathcal{S}} \mathbb{P}\left[\left\|\hat{\theta}_S - \theta_S^*\right\|_{\Sigma_S}^2 \geq t'\right] \\
&\leq \sum_{S \in \mathcal{S}} \delta' \leq \left(\frac{de}{k}\right)^k \delta'
\end{aligned}
$$

Let $\delta = \left(\frac{de}{k}\right)^k \delta'$, and we get $\log(1/\delta') = k\log\frac{d}{k} + k + \log(1/\delta)$. Then, $\sqrt{k} + \sqrt{\log(1/\delta')} \leq 2\sqrt{\log(1/\delta')}$, and

$$
\begin{aligned}
t' &\leq \frac{2\omega^2\sigma^2}{\gamma^2}\frac{\left(2\sqrt{\log(1/\delta')}\right)^2}{n} = \frac{8\omega^2\sigma^2}{\gamma^2}\frac{k\log(d/k) + k + \log(1/\delta)}{n} \\
&\leq \frac{24\omega^2\sigma^2}{\gamma^2}\frac{k\log(d/k) + \log(1/\delta)}{n} =: t
\end{aligned}
$$

Hence,

$$
\begin{aligned}
\mathbb{P}\left[\left\|\hat{\theta}_{\ell_0}^k - \theta^*\right\|_{\Sigma}^2 \geq t\right] &= \mathbb{P}\left[\left\|\left(\hat{\theta}_{\ell_0}^k\right)_{\hat{S}} - \theta_{\hat{S}}^*\right\|_{\Sigma_{\hat{S}}}^2 \geq t\right] \\
&\leq \mathbb{P}\left[\left\|\left(\hat{\theta}_{\ell_0}^k\right)_{\hat{S}} - \theta_{\hat{S}}^*\right\|_{\Sigma_{\hat{S}}}^2 \geq t'\right] \leq \delta
\end{aligned}
$$

### E.3. Proof of Theorem 3.3

For simplicity, we denote $\mathcal{L}(\theta, \{\xi_i\}_{i=1}^n)$ as $\mathcal{L}(\theta)$. By the definition of the $\ell_1$-regularized estimator, we have

$$
\mathcal{L}(\hat{\theta}_{\ell_1}) + \beta\|\hat{\theta}_{\ell_1}\|_1 \leq \mathcal{L}(\theta^*) + \beta\|\theta^*\|_1 \quad \Longleftrightarrow \quad \beta\|\theta^*\|_1 - \beta\|\hat{\theta}_{\ell_1}\|_1 \geq \mathcal{L}(\hat{\theta}_{\ell_1}) - \mathcal{L}(\theta^*)
$$

By the strong log-concavity of $F$ and Lemma E.4, we have

$$
\mathcal{L}(\hat{\theta}_{\ell_1}) - \mathcal{L}(\theta^*) - \left\langle\nabla\mathcal{L}(\theta^*), \hat{\theta}_{\ell_1} - \theta^*\right\rangle \geq \frac{\gamma}{\sigma^2}\|\hat{\theta}_{\ell_1} - \theta^*\|_{\Sigma}^2
$$

Thus,

$$
\begin{aligned}
\frac{\gamma}{\sigma^2}\|\hat{\theta}_{\ell_1} - \theta^*\|_{\Sigma}^2 &\leq \beta\|\theta^*\|_1 - \beta\|\hat{\theta}_{\ell_1}\|_1 - \left\langle\nabla\mathcal{L}(\theta^*), \hat{\theta}_{\ell_1}\right\rangle + \left\langle\nabla\mathcal{L}(\theta^*), \theta^*\right\rangle \\
&\leq \beta\|\theta^*\|_1 - \beta\|\hat{\theta}_{\ell_1}\|_1 + \|\nabla\mathcal{L}(\theta^*)\|_\infty\|\hat{\theta}_{\ell_1}\|_1 + \|\nabla\mathcal{L}(\theta^*)\|_\infty\|\theta^*\|_1 \\
&= (\|\nabla\mathcal{L}(\theta^*)\|_\infty + \beta)\|\theta^*\|_1 + (\|\nabla\mathcal{L}(\theta^*)\|_\infty - \beta)\|\hat{\theta}_{\ell_1}\|_1
\end{aligned}
$$

where the second inequality is by Hölder's inequality. Next, we upper bound $\|\nabla\mathcal{L}(\theta^*)\|_\infty$. We construct a random vector $V = V_S$ as in Lemma E.5 with $S = [d]$, and then $\frac{1}{n\sigma}X_j^\top V = \frac{1}{n\sigma}\sum_{i=1}^n X_{ij}V_i$ is sub-Gaussian with parameter $\frac{\omega^2\|X_j\|_2^2}{n^2\sigma^2} \leq \frac{H^2\omega^2}{n\sigma^2}$ under the assumption that $\max_{1\leq j\leq d}\|X_j\|_2 \leq H\sqrt{n}$. Hence,

$$
\mathbb{P}\left[\|\nabla\mathcal{L}(\theta^*)\|_\infty \geq t\right] = \mathbb{P}\left[\max_{1\leq j\leq d}\frac{1}{n\sigma}|X_j^\top V| \geq t\right] \leq 2d\exp\left(-\frac{t^2 n\sigma^2}{2\omega^2 H^2}\right)
$$

Let $\delta = 2d\exp\left(-\frac{t^2 n\sigma^2}{2\omega^2 H^2}\right)$, and we get $t = \frac{\sqrt{2}\omega H}{\sigma}\sqrt{\frac{\log 2d + \log(1/\delta)}{n}} = \beta$. Thus, with probability at least $1 - \delta$, we have

$$
\begin{aligned}
\frac{\gamma}{\sigma^2}\|\hat{\theta}_{\ell_1} - \theta^*\|_{\Sigma}^2 &\leq (\|\nabla\mathcal{L}(\theta^*)\|_\infty + \beta)\|\theta^*\|_1 + (\|\nabla\mathcal{L}(\theta^*)\|_\infty - \beta)\|\hat{\theta}_{\ell_1}\|_1 \\
&\leq 2\beta\|\theta^*\|_1 \\
\Longrightarrow \|\hat{\theta}_{\ell_1} - \theta^*\|_{\Sigma}^2 &\leq \frac{2\sigma^2}{\gamma}\beta\|\theta^*\|_1
\end{aligned}
$$

### E.4. Proof of Theorem 3.4

For simplicity, we denote $\mathcal{L}(\theta, \{\xi_i\}_{i=1}^n)$ as $\mathcal{L}(\theta)$. By the definition of the $\ell_1$-regularized estimator,

$$\mathcal{L}(\hat{\theta}_{\ell_1}) + \beta\|\hat{\theta}_{\ell_1}\|_1 \leq \mathcal{L}(\theta^*) + \beta\|\theta^*\|_1 \iff \beta\|\theta^*\|_1 - \beta\|\hat{\theta}_{\ell_1}\|_1 \geq \mathcal{L}(\hat{\theta}_{\ell_1}) - \mathcal{L}(\theta^*)$$

By the strong log-concavity of $F$ and Lemma E.4, we have

$$\frac{\gamma}{\sigma^2}\|\hat{\theta}_{\ell_1} - \theta^*\|_\Sigma^2 \leq \mathcal{L}(\hat{\theta}_{\ell_1}) - \mathcal{L}(\theta^*) - \left\langle \nabla\mathcal{L}(\theta^*), \hat{\theta}_{\ell_1} - \theta^* \right\rangle$$

$$\leq \beta\|\theta^*\|_1 - \beta\|\hat{\theta}_{\ell_1}\|_1 - \left\langle \nabla\mathcal{L}(\theta^*), \hat{\theta}_{\ell_1} - \theta^* \right\rangle$$

$$\leq \beta\|\theta^*\|_1 - \beta\|\hat{\theta}_{\ell_1}\|_1 + \|\nabla\mathcal{L}(\theta^*)\|_\infty\|\hat{\theta}_{\ell_1} - \theta^*\|_1$$

where the last inequality is by Hölder's inequality. By assumption, $\|\Sigma - I_d\|_{\max} \leq \frac{1}{32k}$; equivalently, $\left\|\frac{X^\top X}{n} - I_d\right\|_{\max} \leq \frac{1}{32k}$. Hence, $\|X_j\|_2^2 \leq n + 1/(32k) \leq 2n$ for any $j \in [d]$. We construct a random vector $V = V_S$ as in Lemma E.5 with $S = [d]$, and then $\frac{1}{n\sigma}X_j^\top V$ is sub-Gaussian with parameter $\frac{\omega^2\|X_j\|_2^2}{n^2\sigma^2} \leq \frac{2\omega^2}{n\sigma^2}$. Thus,

$$\mathbb{P}\left[\|\nabla\mathcal{L}(\theta^*)\|_\infty \geq t\right] = \mathbb{P}\left[\max_{1 \leq j \leq d}\frac{1}{n\sigma}|X_j^\top V| \geq t\right] \leq 2d\exp\left(-\frac{t^2 n\sigma^2}{4\omega^2}\right)$$

Let $\delta = 2d\exp\left(-\frac{t^2 n\sigma^2}{4\omega^2}\right)$, and then, $t = \frac{2\omega}{\sigma}\sqrt{\frac{\log 2d + \log(1/\delta)}{n}}$. Let $\beta = 2t$, and then, with probability at least $1 - \delta$, it holds that $\|\nabla\mathcal{L}(\theta^*)\|_\infty \leq \frac{\beta}{2}$. Therefore, with probability at least $1 - \delta$,

$$\frac{\gamma}{\sigma^2}\|\hat{\theta}_{\ell_1} - \theta^*\|_\Sigma^2 \leq \beta\|\theta^*\|_1 - \beta\|\hat{\theta}_{\ell_1}\|_1 + \|\nabla\mathcal{L}(\theta^*)\|_\infty\|\hat{\theta}_{\ell_1} - \theta^*\|_1$$

$$\leq \beta\|\theta^*\|_1 - \beta\|\hat{\theta}_{\ell_1}\|_1 + \frac{\beta}{2}\|\hat{\theta}_{\ell_1} - \theta^*\|_1$$

Denote $S := \text{supp}(\theta^*)$, and then,

$$\frac{\gamma}{\sigma^2}\|\hat{\theta}_{\ell_1} - \theta^*\|_\Sigma^2 + \frac{\beta}{2}\|\hat{\theta}_{\ell_1} - \theta^*\|_1 \leq \beta\|\theta^*\|_1 - \beta\|\hat{\theta}_{\ell_1}\|_1 + \beta\|\hat{\theta}_{\ell_1} - \theta^*\|_1$$

$$\leq \beta(\|\theta^*\|_1 - \|\hat{\theta}_{\ell_1}\|_1) + \beta\|(\hat{\theta}_{\ell_1})_S - (\theta^*)_S\|_1 + \beta\|(\hat{\theta}_{\ell_1})_{S^c}\|_1$$

$$= \beta(\|(\theta^*)_S\|_1 - \|(\hat{\theta}_{\ell_1})_S\|_1) + \beta\|(\hat{\theta}_{\ell_1})_S - (\theta^*)_S\|_1$$

$$\leq 2\beta\|(\hat{\theta}_{\ell_1})_S - (\theta^*)_S\|_1$$

where the last inequality is due to $\|(\theta^*)_S\|_1 - \|(\hat{\theta}_{\ell_1})_S\|_1 \leq \|(\hat{\theta}_{\ell_1})_S - (\theta^*)_S\|_1$ by triangle inequality. Since $\frac{\gamma}{\sigma^2}\|\hat{\theta}_{\ell_1} - \theta^*\|_\Sigma^2 \geq 0$, and

$$\frac{\beta}{2}\|\hat{\theta}_{\ell_1} - \theta^*\|_1 = \frac{\beta}{2}\|(\hat{\theta}_{\ell_1})_S - (\theta^*)_S\|_1 + \frac{\beta}{2}\|(\hat{\theta}_{\ell_1})_{S^c}\|_1$$

$$= \frac{\beta}{2}\|(\hat{\theta}_{\ell_1})_S - (\theta^*)_S\|_1 + \frac{\beta}{2}\|(\hat{\theta}_{\ell_1})_{S^c} - (\theta^*)_{S^c}\|_1$$

it holds that

$$\frac{\beta}{2}\|(\hat{\theta}_{\ell_1})_S - (\theta^*)_S\|_1 + \frac{\beta}{2}\|(\hat{\theta}_{\ell_1})_{S^c} - (\theta^*)_{S^c}\|_1 \leq 2\beta\|(\hat{\theta}_{\ell_1})_S - (\theta^*)_S\|_1$$

$$\implies \|(\hat{\theta}_{\ell_1})_{S^c} - (\theta^*)_{S^c}\|_1 \leq 3\|(\hat{\theta}_{\ell_1})_S - (\theta^*)_S\|_1$$

Thus,

$$\|(\hat{\theta}_{\ell_1})_S - (\theta^*)_S\|_1 \leq \sqrt{|S|}\|(\hat{\theta}_{\ell_1})_S - (\theta^*)_S\|_2 \leq \sqrt{k}\|\hat{\theta}_{\ell_1} - \theta^*\|_2 \leq \sqrt{2k}\|\hat{\theta}_{\ell_1} - \theta^*\|_\Sigma$$

where the first inequality is by the Cauchy-Schwarz $\langle a, b \rangle \leq \|a\|_2 \|b\|_2$ by taking $a = \mathbb{1}$, and the last is by Assumption 3.2. Then,

$$\frac{\gamma}{\sigma^2} \|\hat{\theta}_{\ell_1} - \theta^*\|_\Sigma^2 \leq 2\beta \|(\hat{\theta}_{\ell_1})_S - (\theta^*)_S\|_1 \leq 2\beta\sqrt{2k} \|\hat{\theta}_{\ell_1} - \theta^*\|_\Sigma$$

$$\implies \|\hat{\theta}_{\ell_1} - \theta^*\|_\Sigma \leq 2\beta\sqrt{2k} \frac{\sigma^2}{\gamma}$$

$$\implies \|\hat{\theta}_{\ell_1} - \theta^*\|_\Sigma^2 \leq 8\frac{\sigma^4}{\gamma^2}\beta^2 k$$

Again by Assumption 3.2, we have $\|\hat{\theta}_{\ell_1} - \theta^*\|_2^2 \leq 16\frac{\sigma^4}{\gamma^2}\beta^2 k$.

### E.5. Proof of Lemma E.2

The proof of Lemma E.2 has the same idea of the proof of Lemma 8 in Shah et al. (2016) with finite $\mathcal{D}$.

Given $\{(x_{0,i}, x_{1,i})\}_{i=1}^n$, for any $\theta_1, \theta_2 \in \Theta_B \supset \Theta_{B,k}$, the KL divergence between the two distributions $P_{\theta_1}(\{\xi_i\}_{i=1}^n)$ and $P_{\theta_2}(\{\xi_i\}_{i=1}^n)$ of $n$ preference samples with parameters $\theta_1, \theta_2$, respectively, is

$$D_{\text{KL}}(P_{\theta_1}(\{\xi_i\}_{i=1}^n) \parallel P_{\theta_2}(\{\xi_i\}_{i=1}^n)) = \sum_{i=1}^n \left( F\left(\frac{\langle\theta_1, x_{0,i} - x_{1,i}\rangle}{\sigma}\right) \log \frac{F\left(\frac{\langle\theta_1, x_{0,i} - x_{1,i}\rangle}{\sigma}\right)}{F\left(\frac{\langle\theta_2, x_{0,i} - x_{1,i}\rangle}{\sigma}\right)} \right.$$
$$\left. + \left(1 - F\left(\frac{\langle\theta_1, x_{0,i} - x_{1,i}\rangle}{\sigma}\right)\right) \log \frac{1 - F\left(\frac{\langle\theta_1, x_{0,i} - x_{1,i}\rangle}{\sigma}\right)}{1 - F\left(\frac{\langle\theta_2, x_{0,i} - x_{1,i}\rangle}{\sigma}\right)} \right)$$

Since for any $a, b \in (0, 1)$, $a \log \frac{a}{b} \leq (a - b)\frac{a}{b}$, we have

$$D_{\text{KL}}(P_{\theta_1}(\{\xi_i\}_{i=1}^n) \parallel P_{\theta_2}(\{\xi_i\}_{i=1}^n))$$
$$\leq \sum_{i=1}^n \left( \left(F\left(\frac{\langle\theta_1, x_{0,i} - x_{1,i}\rangle}{\sigma}\right) - F\left(\frac{\langle\theta_2, x_{0,i} - x_{1,i}\rangle}{\sigma}\right)\right) \frac{F\left(\frac{\langle\theta_1, x_{0,i} - x_{1,i}\rangle}{\sigma}\right)}{F\left(\frac{\langle\theta_2, x_{0,i} - x_{1,i}\rangle}{\sigma}\right)} \right.$$
$$\left. - \left(F\left(\frac{\langle\theta_1, x_{0,i} - x_{1,i}\rangle}{\sigma}\right) - F\left(\frac{\langle\theta_2, x_{0,i} - x_{1,i}\rangle}{\sigma}\right)\right) \frac{1 - F\left(\frac{\langle\theta_1, x_{0,i} - x_{1,i}\rangle}{\sigma}\right)}{1 - F\left(\frac{\langle\theta_2, x_{0,i} - x_{1,i}\rangle}{\sigma}\right)} \right)$$
$$= \sum_{i=1}^n \frac{\left(F\left(\frac{\langle\theta_1, x_{0,i} - x_{1,i}\rangle}{\sigma}\right) - F\left(\frac{\langle\theta_2, x_{0,i} - x_{1,i}\rangle}{\sigma}\right)\right)^2}{F\left(\frac{\langle\theta_2, x_{0,i} - x_{1,i}\rangle}{\sigma}\right)\left(1 - F\left(\frac{\langle\theta_2, x_{0,i} - x_{1,i}\rangle}{\sigma}\right)\right)}$$

Since $\|\theta_1\|_2, \|\theta_2\|_2 \leq B$, $\|x_{0,i} - x_{1,i}\|_2 \leq L$ for all $i \in [n]$, by the mean value theorem and $\zeta$'s definition (7),

$$D_{\text{KL}}(P_{\theta_1}(\{\xi_i\}_{i=1}^n) \parallel P_{\theta_2}(\{\xi_i\}_{i=1}^n)) \leq \sum_{i=1}^n \zeta \left(\frac{\langle\theta_1, x_{0,i} - x_{1,i}\rangle}{\sigma} - \frac{\langle\theta_2, x_{0,i} - x_{1,i}\rangle}{\sigma}\right)^2$$
$$= \frac{n\zeta}{\sigma^2}(\theta_1 - \theta_2)^\top \Sigma (\theta_1 - \theta_2)$$
$$= \frac{n\zeta}{\sigma^2} \|\theta_1 - \theta_2\|_\Sigma^2$$

## E.6. Proof of Lemma E.4

The gradient of $\ell_S$ is of the form

$$(\nabla\ell_S(\theta))_S = -\frac{1}{n\sigma}\sum_{i=1}^n \left(\mathbb{1}_{\{y_i=0\}}\cdot\frac{F'\left(\langle\theta_S,(x_{0,i}-x_{1,i})_S\rangle/\sigma\right)}{F\left(\langle\theta_S,(x_{0,i}-x_{1,i})_S\rangle/\sigma\right)}\right.$$
$$\left.+\mathbb{1}_{\{y_i=1\}}\cdot\frac{-F'\left(\langle\theta_S,(x_{0,i}-x_{1,i})_S\rangle/\sigma\right)}{1-F\left(\langle\theta_S,(x_{0,i}-x_{1,i})_S\rangle/\sigma\right)}\right)(x_{0,i}-x_{1,i})_S$$

$$(\nabla\ell_S(\theta))_{S^c}=0$$

The sub-matrix of the Hessian of $\ell_S$ indexed by $S\times S$ is

$$\left(\nabla^2\mathcal{L}(\theta)\right)_{SS}=\frac{1}{n\sigma^2}\sum_{i=1}^n\left(\mathbb{1}_{\{y_i=0\}}\cdot T_{0,i}+\mathbb{1}_{\{y_i=1\}}\cdot T_{1,i}\right)(x_{0,i}-x_{1,i})_S(x_{0,i}-x_{1,i})_S^\top$$

where

$$T_{0,i}=\frac{(F'\left(\langle\theta_S,(x_{0,i}-x_{1,i})_S\rangle/\sigma\right))^2-F\left(\langle\theta_S,(x_{0,i}-x_{1,i})_S\rangle/\sigma\right)F''\left(\langle\theta_S,(x_{0,i}-x_{1,i})_S\rangle/\sigma\right)}{(F\left(\langle\theta_S,(x_{0,i}-x_{1,i})_S\rangle/\sigma\right))^2}$$
$$=\nabla^2\log F\left(\langle\theta_S,(x_{0,i}-x_{1,i})_S\rangle/\sigma\right)\geq 2\gamma$$
$$T_{1,i}=\frac{(F'\left(\langle\theta_S,(x_{0,i}-x_{1,i})_S\rangle/\sigma\right))^2+(1-F\left(\langle\theta_S,(x_{0,i}-x_{1,i})_S\rangle/\sigma\right))F''\left(\langle\theta_S,(x_{0,i}-x_{1,i})_S\rangle/\sigma\right)}{(1-F\left(\langle\theta_S,(x_{0,i}-x_{1,i})_S\rangle/\sigma\right))^2}$$
$$=\nabla^2\log(1-F\left(\langle\theta_S,(x_{0,i}-x_{1,i})_S\rangle/\sigma\right))$$
$$=\nabla^2\log(F\left(-\langle\theta_S,(x_{0,i}-x_{1,i})_S\rangle/\sigma\right))\geq 2\gamma$$

and zero at the entries in the complement of $S\times S$. Hence, for any $v\in\mathbb{R}^d$, we have for any $\theta\in\Theta_B$,

$$v^\top\nabla^2\mathcal{L}(\theta)v=v_S^\top\left(\nabla^2\mathcal{L}(\theta)\right)_{SS}v_S\geq\frac{2\gamma}{\sigma^2}\|v_S\|_{\Sigma_S}^2$$

Thus, for any $\theta'\in\mathbb{R}^d$ such that $\theta^*+\theta'\in\Theta_B$, it holds that

$$\ell_S(\theta^*+\theta')-\ell_S(\theta^*)-\langle\nabla\ell_S(\theta^*),\theta'\rangle\geq\frac{\gamma}{\sigma^2}\|\theta'_S\|_{\Sigma_S}^2$$

## E.7. Proof of Lemma E.5

$$(\nabla\ell_S(\theta^*))_S=-\frac{1}{n\sigma}\sum_{i=1}^n\left(\mathbb{1}_{\{y_i=0\}}\cdot\frac{F'\left(\langle\theta_S^*,(x_{0,i}-x_{1,i})_S\rangle/\sigma\right)}{F\left(\langle\theta_S^*,(x_{0,i}-x_{1,i})_S\rangle/\sigma\right)}\right.$$
$$\left.+\mathbb{1}_{\{y_i=1\}}\cdot\frac{-F'\left(\langle\theta_S^*,(x_{0,i}-x_{1,i})_S\rangle/\sigma\right)}{1-F\left(\langle\theta_S^*,(x_{0,i}-x_{1,i})_S\rangle/\sigma\right)}\right)(x_{0,i}-x_{1,i})_S$$

Define $V_S$ as

$$(V_S)_i:=\begin{cases}\frac{F'\left(\langle\theta_S^*,(x_{0,i}-x_{1,i})_S\rangle/\sigma\right)}{F\left(\langle\theta_S^*,(x_{0,i}-x_{1,i})_S\rangle/\sigma\right)}, & \text{w.p. } F\left(\langle\theta_S^*,(x_{0,i}-x_{1,i})_S\rangle/\sigma\right)\\\frac{-F'\left(\langle\theta_S^*,(x_{0,i}-x_{1,i})_S\rangle/\sigma\right)}{1-F\left(\langle\theta_S^*,(x_{0,i}-x_{1,i})_S\rangle/\sigma\right)}, & \text{w.p. } 1-F\left(\langle\theta_S^*,(x_{0,i}-x_{1,i})_S\rangle/\sigma\right)\end{cases}$$

Then, $(\nabla\ell_S(\theta^*))_S=-\frac{1}{n\sigma}X_S^\top V_S$. Notice that $\mathbb{E}[V_S]=0$, and by the definition (7) of $\omega$,

$$|(V_S)_i|\leq\sup_{z\in[-2BL/\sigma,2BL/\sigma]}\left\{\frac{F'(z)}{F(z)},\frac{F'(z)}{1-F(z)}\right\}=\omega$$

