# OpenReview forum: "Leveraging Sparsity for Sample-Efficient Preference Learning: A Theoretical Perspective"
_ICML.cc/2025/Conference — ICML 2025 poster_

### Official Review · Reviewer_Q6xK · 2025-03-01

**Overall Recommendation:** 3

**Summary:**

The paper explores the impact of sparsity in preference learning, establishing a minimax lower bound on empirical error under sparse RUM and deriving upper bounds for two sparsity-regularized estimators. The experiments, conducted on both a synthetic dataset and an LLM alignment setting, validate the theoretical findings.

**Claims And Evidence:**

Yes.

**Essential References Not Discussed:**

Reference well discussed.

**Experimental Designs Or Analyses:**

Yes.

**Methods And Evaluation Criteria:**

Yes.

**Other Comments Or Suggestions:**

No.

**Other Strengths And Weaknesses:**

**Strengths:**
- The paper is comprehensive, establishing both a lower bound and upper bounds for different estimators, along with a clear comparison between them.
- The experimental setup is well-designed and effectively validates the theoretical contributions.
- The writing is clear and seamlessly integrates the two topics in a straightforward manner.

**Weaknesses:**
- It focuses solely on the fixed design setup (i.e., a deterministic set of pairs) and does not address generalization error.

**Questions For Authors:**

- To address the limitation of the fixed design setup, can we explore ways to generalize the model?
- The experimental results primarily focus on the accuracy of the reward model. Can we take this further by evaluating LLMs after applying PPO with the sparsity-regularized reward model?
- Maybe I missed the details, in Appendix D.2, you report accuracy versus the number of samples. How are these samples selected? Are they guaranteed to be the same set for training with and without L1 regularization?
- Just brainstorm, to verify the presence of sparsity in LLMs, could we train a reward model without regularization at different sample sizes, measure its sparsity, and analyze whether sparsity naturally increases as training progresses?

**Relation To Broader Scientific Literature:**

The paper connects the well-known role of sparsity in traditional compressed sensing to preference learning, highlighting its ubiquity across broader literature.

**Theoretical Claims:**

I did not check the details into the proofs due to time constraints.

---

> ### Author Rebuttal · Authors · 2025-03-30
>
> We thank the reviewer for the thoughtful and constructive feedback. We are glad that the reviewer found the theoretical contributions comprehensive, the experiments well-designed, and the writing clear in presenting the connections between sparsity and preference learning. Below, we respond to each of the comments in detail.
>
> (1) *"It focuses solely on the fixed design setup (i.e., a deterministic set of pairs) and does not address generalization error." "To address the limitation of the fixed design setup, can we explore ways to generalize the model?"*
>
> We appreciate the reviewer for this comment. While our current study focuses on a fixed design setup, this setting is commonly encountered in real-world applications, such as crowdsourced comparisons or benchmark datasets, where the pairwise comparisons are predetermined.  That said, extending our analysis to the random design setup and studying generalization error is an exciting and important direction for future work. We thank the reviewer for highlighting this point. In particular, recent work on linear regression under random design (e.g., [1-3]) may provide valuable insights. For example, as suggested in [2], techniques from random matrix theory and random projections may be helpful for such an extension.
>
> [1] https://jmlr.org/papers/volume23/19-571/19-571.pdf
>
> [2] https://arxiv.org/pdf/2303.01372
>
> [3] https://arxiv.org/pdf/2203.08564
>
> (2) *"The experimental results primarily focus on the accuracy of the reward model. Can we take this further by evaluating LLMs after applying PPO with the sparsity-regularized reward model?"*
>
> We thank the reviewer for this suggestion. Indeed, a natural extension is to evaluate LLMs fine-tuned via PPO using the learned reward model. For example, given a set of prompts, one could compare LLM-generated responses through human evaluation or an oracle model, using  win rate as a downstream metric. Although we initially considered such a metric, we eventually chose not to pursue it in this paper to maintain our focus on reward modeling without incorporating additional policy optimization. Nonetheless, we agree with the reviewer that this evaluation approach is a compelling direction for future work.
>
> (3) *"Maybe I missed the details, in Appendix D.2, you report accuracy versus the number of samples. How are these samples selected? Are they guaranteed to be the same set for training with and without L1 regularization?"*
>
> Yes, we use the same set of samples across models to ensure a fair comparison. Specifically, for each trial, the training data is sampled uniformly at random and fixed in advance; this identical dataset is then used to train both the regularized and unregularized models. We will make this point more explicitly in the revised version.
>
> (4) *"Just brainstorm, to verify the presence of sparsity in LLMs, could we train a reward model without regularization at different sample sizes, measure its sparsity, and analyze whether sparsity naturally increases as training progresses?"*
>
> We appreciate this interesting idea. In theory, the estimation error in $\ell_2$-norm $\lVert\hat{\theta} - \theta^*\rVert_2$ without sparse-regularization will gradually be small given that 1) the number of samples is sufficiently large relative to the feature dimension $d$, and 2) the Gram matrix $\Sigma$ is well-behaved, e.g., satisfying the restricted eigenvalue condition. As a result, in practice, if the ground-truth parameter $\theta^*$ is sparse, the learned parameter would be sparse even without regularization under the above conditions. Nonetheless, given the high dimensionality of the feature space in LLM-based reward models, the current dataset does not meet the scale required for such behavior to emerge. We appreciate the suggestion and agree that this would be an interesting empirical direction to pursue with enough annotated data.
>
> We once again thank the reviewer for the recognition and thoughtful questions. We believe these discussions will help guide valuable extensions of this work.

---

### Official Review · Reviewer_PinM · 2025-03-11

**Overall Recommendation:** 4

**Summary:**

This paper proposes a sparse setting for preference learning, the authors state that human preferences are driven by some critial factors, of which the dimension is generally low.
Therefore, the authors study the preference learning problem in the sparse setting from a theoretical perspective, deriving bounds for estimation error under $l_0$ and $l_1$ regularizations, respectively.

**Claims And Evidence:**

I think the k-sparse claim for RUM needs further evidence.
Although the authors provide better theoretic bounds for estimation error in dimension $d$, the overall claim of this paper that RUM is sparse is not validated in experiments.
The first experiment directly uses this assumption to sample the ground-truth $\theta^*$, and the second experiment uses the prediction accuracy as the evaluation metric, which is supportive but not directly linked to the sparsity assumption.
I wonder if there're other baselines the authors could use in the literature to support this assumption.

I have no problems with other assumptions.

**Essential References Not Discussed:**

I think the authors have covered the most related literature.

**Experimental Designs Or Analyses:**

The experiment 4.1 validates the effectiveness of $l_1$ regularization compared with standard MLE in the preference learning case.
However, the ground-truth parameter $\theta^*$ is manually set to be sparse, which undermines this experiment.
Also, this experiment does not justify the assumption that RUM in preference learning satisfies sparsity.

The experiment 4.2 uses LLM alignment as the task, and the reward model accuracy on the test set is used as the evaluation metric.
However, the rm-static dataset, which is a split of HH dataset [1], may contain noise [2], reducing the reliability of the experiment results.
The authors should consider adding a cleaner dataset for experiment.

[1] https://huggingface.co/datasets/Anthropic/hh-rlhf

[2] Impact of preference noise on the alignment performance of generative language models, COLM 2024

**Methods And Evaluation Criteria:**

If the authors could have further justification for the sparsity assumption, then the proposed methods make sense.

Actually, the second experiment could be conducted with one more dataset, to provide more comprehensive evaluation.

**Other Comments Or Suggestions:**

From a reader's perspective, I would suggest the authors adding references for theorem derivations around each theorem in the main content.

**Other Strengths And Weaknesses:**

Strengths:

The theoretical derivations in this paper are complete and provide insights for learning utility functions based on pairwise comparisons under the sparsity assumption.

Weaknesses:
1. The sparsity assumption in this paper is not well justified.
2. The experiments lack sufficient baselines.

**Questions For Authors:**

1. How would the authors justify that using sparsity in this scenario is a reasonable approach?
2. Is the experiment in section 4.1 necessary? As previous research has shown that $l_1$ regularization works well under the sparsity condition in MLE, what is the main difference between preference learning under sparsity and previous literature?
3. Are there any other evaluation metrics for the quality of reward models in experiment 4.2?

**Relation To Broader Scientific Literature:**

The authors may consider apply the sparsity setting to other preference learning algorithms with utility models, for example, DPO [1];
The sparsity assumption might somehow relate to sparse attention [2, 3], but in different perspectives.


[1] Direct Preference Optimization: Your Language Model is Secretly a Reward Model, NeurIPS 2023

[2] Linformer: Self-Attention with Linear Complexity

[3] Efficient Streaming Language Models with Attention Sinks, ICLR 2024

**Theoretical Claims:**

Yes, I checked the derivation of the four main theorems in the Appendix.

---

> ### Author Rebuttal · Authors · 2025-03-30
>
> We thank the reviewer for the thoughtful and constructive feedback. We greatly appreciate the recognition of our theoretical contributions and the time taken to carefully examine the proofs of our four main theorems. Below, we address the comments and questions in detail.
>
> **(1) Evidence for k-sparse assumption for RUM.** *"I think the k-sparse claim for RUM needs further evidence…"*
>
> We thank the reviewer for raising this point, which was also noted by **Reviewer ACSW**. A detailed response is provided there in **(2) On the sparsity assumption and its practical validity**.  Due to space constraints, we refer the reviewer to that response. Here, we offer additional clarifications specific to the reviewer’s comments.
>
> - *"I wonder if there're other baselines the authors could use in the literature to support this assumption…"*
>
>     As noted in the introduction, the theoretical and empirical foundations for sparsity in preference learning remain largely underexplored. Nevertheless, we observe increasing interest in related domains. For example, a very recent work by Jin et al. (Sparsity-Agnostic Linear Bandits with Adaptive Adversaries, NeurIPS 2024) studies sparsity in linear reward functions in the context of adaptive bandits. While not directly addressing pairwise comparisons, this work provides complementary motivation for sparse reward structures.
>
> - *"the second experiment uses the prediction accuracy as the evaluation metric, which is supportive but not directly linked to the sparsity assumption"*
>
>     We thank the reviewer for the constructive feedback. Empirically, we observe that the learned parameters under $\ell_1$ regularization are indeed highly sparse. For example, in Figure 4 (frozen backbone setting), the learned sparsity level $k/d$ is around 4-8% for both datasets, and the sparsity-aware method consistently outperforms the baseline. These results help to empirically validate the sparsity assumption. We will include these results in the updated version.
>
> **(2) Real-Data Experiments.**
> - *"the second experiment could be conducted with one more dataset…" "The experiments lack sufficient baselines."*
>
>     We thank the reviewer for this suggestion. In addition to rm-static, we include results on the SHP dataset in Appendix D due to page limitation. Notably, SHP contains human-written responses, while rm-static contains machine-generated ones, giving us two distinct distributions, as mentioned by Ethayarajh et al., (2022). In both cases, $\ell_1$ regularization outperforms the baseline.
> - *"However, the rm-static dataset… may contain noise [2], reducing the reliability of the experiment results. The authors should consider adding a cleaner dataset for experiment."*
>
>     We thank the reviewer for the observation. While rm-static is widely used as a benchmark dataset for alignment-based preference learning, we agree that evaluating on cleaner datasets, or applying data-filtering techniques (e.g., confidence-based data filtering in [2]) beforehand, could provide additional value. We are very interested exploring this direction and will consider it in the future work.
> - *"Are there any other evaluation metrics for the quality of reward models in experiment 4.2?"*
>
>     We thank the reviewer for this question. One indirect way is to evaluate the performance of an aligned LLM associated with the reward models. Specifically, one can sample prompts, generate responses from the aligned LLMs, and then assess quality via human annotation or an oracle model. The win rate of the aligned LLM can serve as a downstream measure of the quality of the  reward model.
>
> **(3) Synthetic Experiments.** *"The experiment 4.1 validates the effectiveness of l1-regularization compared with standard MLE in the preference learning case. However, the ground-truth is manually set to be sparse, which undermines this experiment." "Is the experiment in section 4.1 necessary? … what is the main difference between preference learning under sparsity and previous literature?"*
>
> We thank the reviewer for this question. As correctly pointed out, the purpose of Experiment 4.1 is not to justify the sparsity assumption, but to validate the effectiveness of $\ell_1$ regularization in the sparse RUM setting. It serves as a sanity check for our theoretical analysis, which provides estimation error bounds under sparse ground truth. While $\ell_1$ regularization has been well-studied in sparse regression and related settings, sparse preference learning under RUM poses unique challenges. In particular, each pair of data only provides one-bit comparison feedback, rather than direct access to real-valued rewards. For completeness, we include the numerical experimental results in Section 4.1 in the paper.
>
> We thank the reviewer again for the constructive comments and the recognition of our work. We will incorporate the above clarifications, along with other suggestions from the reviewer, in our updated version.

---

> > ### Comment · Reviewer_PinM · 2025-04-02
> >
> > I appreciate the authors' response to my questions. Most of my concerns regarding the justification of sparsity assumption, and the soundness of experiments are resolved.
> > Based on the overall quality of the work, I decide to maintain my score.

---

### Official Review · Reviewer_3Ass · 2025-03-11

**Overall Recommendation:** 4

**Summary:**

The authors analyze the sample complexity of RUMs where utilities are an inner product between $d$-dimensional item features $x$ and preference parameters $\theta$, and where $\theta$ is $k$-sparse. Whereas existing results on the sample complexity of learning $\theta$ find that error decays as $\Theta(d / n)$ with $n$ samples, the authors show that when $\theta$ is $k$-sparse, this rate can be improved to $\Theta(k \log (d / k) / n)$, a large improvement when $d$ is large and $k$ is small. Moreover, the authors show that the $\ell_1$-regularized MLE has a near-optimal sample complexity. In experiments on synthetic data, the authors demonstrate the sample complexity benefit of using $\ell_1$ regularization, in line with their theoretical results. They also demonstrate the usefulness of accounting for the sparsity of preferences with $\ell_1$ regularization in RLHF of LLMs.

### Update after rebuttal
Ah, the squared vs not squared norm explains my confusion. Thanks! Glad to see the other reviewers agree on recommending acceptance.

**Claims And Evidence:**

All claims are well supported by theoretical, simulation, and experimental evidence.

**Essential References Not Discussed:**

There is another ICML paper that comes to mind involving sparse RUMs (where only a subset of features are "salient" in a particular comparison) that also derives sample complexity rates:
Bower & Balzano. Preference Modeling with Context-Dependent Salient Features, ICML 2020:
https://proceedings.mlr.press/v119/bower20a.html

From what I can tell, their convergence rate is $O(\sqrt{d \log d / n})$. I think this paper is very much worth discussing in relation to authors' results.


A few other papers that could have been in the related work, but are not essential (listing in case the authors were not aware of these and feel that any of them are worth including)
- https://proceedings.mlr.press/v97/seshadri19a.html (has sample complexity results for a RUM variant)
- https://proceedings.neurips.cc/paper/2020/hash/6affee954d76859baa2800e1c49e2c5d-Abstract.html (above model applied to rankings, in the style of Plackett-Luce; also has sample complexity results)
- https://dl.acm.org/doi/10.1145/3447548.3467250 (uses $\ell_1$ regularization in a RUM to learn sparse parameters)
- https://doi.org/10.1017/nws.2023.20 (discusses sample complexity benefit of Laplacian regularization for RUMs with preferences correlated over graphs)
- https://proceedings.mlr.press/v119/rosenfeld20a.html (another ML-based RUM variant)

**Experimental Designs Or Analyses:**

The experiments and subsequent analyses, while not central to the paper, are well done.

**Methods And Evaluation Criteria:**

The methods and evaluation criteria are solid.

**Other Comments Or Suggestions:**

1. In 1.2, it's worth mentioning in the contribution bullet points that the $\ell_0$-regularized estimator is infeasible in practice, but $\ell_1$-regularization is practical (otherwise from reading this section, it's not clear why we wouldn't always use $\ell_0$ regularization). This point is made very clearly in 3.2.1 and 3.2.2.

**Other Strengths And Weaknesses:**

I wish all papers were this well written; thanks to the authors for submitting something so polished. The results and experiments are very nice and I would like to see this paper accepted. It just needs a discussion of Bower & Balzano (2020) and I would like to resolve my question below about the convergence rate from Zhu et al (2023).


Strengths:
- The paper is extremely clearly written, to a very high standard.
- The problem is both timely in its applications to RLHF and timeless in its application to choice modeling more broadly
- The results appear to be a significant improvement on prior sample complexity results in the sparse setting, which is very encouraging for using $\ell_1$ regularization
- The experiments do a great job of succinctly demonstrating the benefit of leveraging sparsity through $\ell_1$ regularization

Weaknesses:
- People have been using $\ell_1$ regularization when fitting choice models with sparse parameters for many years, so while these new bounds provide some nice theoretical grounding for this approach, it doesn't necessarily impact practice. Perhaps raising this issue is necessary for the RLHF crowd
- One important related work on RUMs with sparse features is not discussed

**Questions For Authors:**

1. The bounds in Zhu, Jodan, and Jiao 2023 appear to be $O(\sqrt{d / n})$ rather than $O(d / n)$ as stated Table 1. Is there something I'm missing about their results?
2. How does this model and these sample complexity results compare to those in Bower & Balzano (2020)?

**Relation To Broader Scientific Literature:**

This paper provides theoretical backing for the practice of $\ell_1$ regularization in RUMs when the utility parameters are sparse, proving an asymptotic improvement of the best known sample complexity rates when preferences are sparse.

**Theoretical Claims:**

The claims are very clearly stated and the supplement has proofs. I skimmed the proof of Theorem 3.1, which looks reasonable.

---

> ### Author Rebuttal · Authors · 2025-03-30
>
> We appreciate the reviewer’s recognition of the clarity of our presentation, the significance of the theoretical contributions, and the relevance of our work to both RLHF and choice modeling. We also thank the reviewer for taking the time to examine the proofs, and are glad that the experimental results were found solid. We address the reviewer’s comments and questions below.
>
> **(1) Comparison with Bower & Balzano (2020).** *"One important related work on RUMs with sparse features is not discussed." "How does this model and these sample complexity results compare to those in Bower & Balzano (2020)?"*
>
> We thank the reviewer for pointing out this closely related paper. While both works consider comparisons based on a subset of features, there are two key differences: 1) In our setting, the subset of relevant features, which correspond to nonzero entries in a globally sparse parameter vector, is fixed across all comparisons. In contrast, Bower & Balzano (2020) identify salient features in a context-dependent manner, selecting those with the largest pairwise variance for each comparison. 2) Our subset of features are selected through optimization whereas Bower & Balzano (2020) select feature sets through sample variance of features. As a result, their model is not globally sparse, and the corresponding estimation rate is $O(d \log (d))$, whereas our analysis yields a sharper rate of $O(k \log (d/k))$ under global sparsity.
>
> **(2) Clarifying the rate from Zhu et al. (2023).** *"The bounds in Zhu, Jordan, and Jiao 2023 appear to be rather than as stated Table 1. Is there something I'm missing about their results?"*
>
> We thank the reviewer for this observation. We note that our results are stated in terms of the **squared semi-norm** $\lVert \cdot \rVert_\Sigma^2$ as the error metric, whereas Zhu et al. (2023) report bounds in terms of the **semi-norm** $\lVert \cdot \rVert_\Sigma$. We will clarify this in the caption of Table 1 to avoid confusion.
>
> **(3) Brief review of the listed works.** *“A few other papers that could have been in the related work, but are not essential (listing in case the authors were not aware of these and feel that any of them are worth including)”*
>
> We thank the reviewer for highlighting these additional related works and pointing out their connection to our work. We will include the references and discussion in the updated version of the paper.
>
> We will incorporate the above discussion, along with the other suggestions mentioned by the reviewer, in our updated version of the paper.

---

### Official Review · Reviewer_ACSW · 2025-03-11

**Overall Recommendation:** 3

**Summary:**

The paper investigates leveraging sparsity in preference learning to achieve improved sample efficiency. Under the sparse random utility model (RUM), the authors derive minimax optimal estimation rates, emphasizing the theoretical benchmark provided by an $l_0$-constrained estimator. However, recognizing that this estimator is computationally intractable, the authors propose practical $l_1$-regularized estimators and rigorously establish their estimation guarantees. Empirical evaluations substantiate that the $l_1$-regularized estimator, being computationally feasible, achieves significant improvements over standard methods, particularly when sparsity is present.

**Claims And Evidence:**

1. When the sparsity is present, *sparse preference learning* reduces the sample complexity from $\Theta(d/n)$ to $\Theta(k/n \log(d/k))$. The claim is well supported by theoretical derivations. The statements about matching upper and lower bounds are accompanied by detailed proofs.

2. An $l_1$-regularized estimator can be made computationally tractable. Under an additional assumption, it can achieve a near-optimal rate. In particular, a "fast rate" of $O(k/n\log(d))$ is shown if the Gram matrix satisfies a restricted eigenvalue-type condition. The claims are also well-supported.

**Essential References Not Discussed:**

Nothing noteworthy to me.

**Experimental Designs Or Analyses:**

In Section D.1, they try full fine-tuning (rather than just the last layer) on the same dataset and metrics to see if regularization still helps. The setup seems solid, with consistent hyperparameter choices and a straightforward accuracy measure. However, the improvement they report is only about 3%, which isn’t huge. It raises the question of whether the extra effort is worth such a small gain. Even so, I don’t see any major flaws in how they designed or evaluated these experiments.

**Methods And Evaluation Criteria:**

1. I think $l_0$-constrained MLE method is mainly for the theoretical purpose. It makes sense to develop $l_1$-regularized MLE method, both theoretically and empirically.

2. The authors use synthetic data to control sparsity and dimension, and they verify that the $l_1$-based estimator outperforms the unregularized baseline in small-sample, large-dimension conditions, which is consistent with the theoretical predictions.

**Other Comments Or Suggestions:**

It might be useful for you to study widely used real-world reward models—perhaps from different RLHF or recommendation pipelines—and check if they already show sparsity in practice. If they do, your approach is strongly reinforced; if not, you could explore alternative ways to handle approximate or partial sparsity.

**Other Strengths And Weaknesses:**

Strengths:
1. Their theoretical analysis is thorough and lines up with known compressed sensing results.
2. They give a clear reason for using sparse preference learning. I think the direction is novel and meaningful.

Weakness:
1. It is unknown that how often the ground-truth parameter is sparse in real-world application. Hence, it is unclear how effective the approach actually is.
2. The reported performance gains can be modest, which might limit its practical impact.

**Questions For Authors:**

1. Can you provide insights or examples of practical scenarios in which your assumptions are realistically met or potentially violated?
2. Can you share a few insights about the implication of your work on direct preference optimization (DPO)?

**Relation To Broader Scientific Literature:**

The paper fits into the broader RLHF literature by addressing sample-efficiency concerns in reward modeling.

**Theoretical Claims:**

I did not check the proofs in detail.

---

> ### Author Rebuttal · Authors · 2025-03-30
>
> We thank the reviewer for the thoughtful feedback and the recognition of our theoretical contributions, the clarity of our motivation, the novelty of applying sparsity to preference learning, and the solid experimental setup. Below, we address the concerns and questions:
>
> **(1) On the empirical gains (~3%).**  *“The improvement they report is only about 3%, which isn’t huge. It raises the question of whether the extra effort is worth such a small gain”*
>
> - **Limited room for improvement due to inherent randomness.**  Under RUM, the probability of a pairwise preference is given by $P(A\succ B) = F(\frac{r^*(A) - r^*(B)}{\sigma})$, which captures the inherent randomness in human decision-making. Thus, the same pair (A, B) may receive conflicting annotations, **fundamentally bounding the maximum achievable test accuracy**. For example, if 7 out of 10 data points prefer $A\succ B$, and 3 prefer $B\succ A$, then the maximum achievable accuracy on this pair is 70%. In real-world datasets, [1] reports 19–37% of crowd-labeled preferences are noisy. This implies that even with the ground-truth model, the expected accuracy would be limited to roughly 63–81%. Given that a random guess yields 50% accuracy, a 3% improvement represents a substantial fraction of the remaining improvement room.
> - **Consistent gains and interpretability with negligible extra effort.** Adding regularization term to the training loss does not change the model architecture and incurs marginal overhead. Despite its simplicity, $\ell_1$ regularization consistently improves performance, particularly in low-sample regimes. Moreover, the sparsity induced by $\ell_1$ regularization enhances interpretability by identifying the most relevant features for the preference model of interest.
>
> **(2) On the sparsity assumption and its practical validity.** *“It is unknown how often the ground-truth parameter is sparse in real-world application. Hence, it is unclear how effective the approach actually is.”  “Can you provide insights or examples when your assumptions are realistically met or potentially violated?”*
>
> - **When does sparsity arise?** Sparsity commonly emerges in classical preference learning, especially in high-dimensional feature space where human preferences are influenced by only a few factors. For example, a user’s preference for smartphones may depend primarily on price, camera quality, and UI design, whereas many other attributes (e.g., place of manufacture) may have little influence for that user. Similarly, a reader’s preference over articles may depend solely on the presence of a few key words. When the feature vector is a binary indicator over a large vocabulary, the reward parameter is naturally sparse.
> - **When might the assumption be violated?** Sparsity may not hold when the feature space is low-dimensional (e.g., 3–5 features), or when features have been manually curated. In such cases, the benefits of sparsity-aware estimation are expected to diminish. Nonetheless, in many modern applications, such as LLM alignment or recommendation systems, feature spaces often contain thousands to millions of dimensions. In these settings, identifying relevant features beforehand is infeasible, making sparsity both a practically and necessary modeling assumption.
> - **Empirical support in LLM alignment.** Our LLM alignment experiments show that $\ell_1$ regularization induces highly sparse models while outperforming the baseline, even without hyperparameter tuning. For example, in the frozen backbone training setup (Figure 4, Section D.2), where the pre-trained model defines the feature map $\phi$, we observe:
>     - rm-static dataset: $k/d\approx$ 4.5% (n=800) and $k/d\approx$ 7.5% (n=3200).
>     - SHP dataset: $k/d\approx$ 4.2% (n=800) and $k/d\approx$ 7.2% (n=3200).
>
>     These results show that $\ell_1$ regularization selects a small and informative subset of features, reinforcing the practical validity of the sparsity assumption.
>
> **(3) Implication on DPO.**  *“Can you share a few insights about the implication of your work on DPO?”*
>
> DPO ([2]) bypasses explicit reward modeling by directly optimizing the policy, where $\beta \log(\pi/\pi_{\text{ref}})+\beta\log Z$ acts as a proxy reward that depends on the policy $\pi$. This contrasts with reward-based approaches, where reward and policy are decoupled, allowing assumptions (e.g., sparsity) and regularization to be applied directly to the reward model. However, since DPO couples the reward signal with the policy and the policy architecture is fixed, incorporating sparsity-aware regularization is non-trivial. Additionally, even when the reward model $r(x)$ is sparse, the parameter of the optimal policy $\pi^*(x) = \frac{1}{Z}\pi_{\text{ref}}(x)\exp(\frac{1}{\beta}r(x))$ is not necessarily sparse.
>
> We thank the reviewer again for the insightful comments. We will incorporate the discussion in the updated version.
>
> [1]https://arxiv.org/pdf/2306.05685
>
> [2]https://arxiv.org/pdf/2305.18290

---

### Decision · Program_Chairs · 2025-05-01

**Decision:**

Accept (poster)

**Comment:**

Paper studies choice modeling (preference learning) under $k$-sparsity constraint on the $d$-dimensional parameters. Authors show that if the true model is sparse then the minimax sample complexity lower-bound for learning the choice model is only $\widetilde{\Omega}(k)$ instead $\widetilde{\Omega}(d)$. Further, they show that $\ell_1$ regularized MLE estimator achieves this rate. Finally, paper uses synthetic experiments and very few LLM reward modeling results to showcase their algorithm. The paper is well written and the proofs appear correct. However, it was noted that analysis and algorithmic techniques used in the paper mostly follow from past literature on compressed sensing. So overall the theoretical and synthetic results are not surprising. Further, since the paper studies choice modeling in the context of RLHF, authors could have spent more effort on evaluating and verifying their ideas on more reward modeling datasets and LLMs and its effectiveness.